**Prediction of volume of shallow landslides due to rainfall using data-driven models**

Tuganishuri Jérémie[1], Chan-Young Yune[2], Gihong Kim[3], Seung Woo Lee[4], Manik Das Adhikari[5], Sang-GukYum[6*]

 Department of Civil and Environmental Engineering, Gangneung-Wonju National University,

*Corresponding author: Sang-Guk Yum; skyeom0401@gwnu.ac.kr

**Abstract**

Landslides due to rainfall are among the most destructive natural disasters that cause property damages, huge financial losses, and human deaths in different parts of the World. To plan for mitigation and resilience, the prediction of the volume of rainfall-induced landslides is essential to understand the relationship between the volume of soil materials debris and their associated predictors. Objectives of this research are to construct a model by utilizing advanced data-driven algorithms (i.e., ordinary least square or Linear regression (OLS), random forest (RF), support vector machine (SVM), extreme gradient boosting (EGB), generalized linear model (GLM), decision tree (DT), and deep neural network (DNN), K-nearest neighbor (KNN) and Ridge regression (RR)) for the prediction of the volume of landslides due to rainfall considering geological, geomorphological, and environmental conditions. Models were tested on the Korean landslide dataset to ~~observe~~obtain the ~~best performing model, and among tested algorithms, the~~ most efficient predictions. The extreme gradient boosting ~~ranked high with~~predictions exhibited the highest coefficient of determination ($R^2$=0.~~858~~841) and lowest mean absolute error (MAE=~~150.421m$^3$~~146.6120 m$^3$), followed by random forest ($R^2$=0.8435, MAE=330.4876 m$^3$). The volume of landslides was strongly influenced by slope length, ~~drainage status~~maximum hourly rainfall, slope angle, aspect, and ~~age of trees~~altitude. The anticipated volume of ~~landslide~~landslides can be important for land use allocation and efficient landslide risk management.

*Keywords:* Data-driven models, volume of ~~landslide~~landslides, prediction models, rainfall, South Korea

## 1. Introduction

Landslides due to rainfall ~~is~~are phenomena that dislocate a ~~phenomenon in which a given volume~~mass of soil ~~dislocates~~ from its ~~original high to lower point altitude~~ natural position and slide downward along a slope due to gravity forces ~~along a slope fragilized by~~. Intense or long-duration rainfall ~~that~~infiltrates the soil and increases the pore pressure, resulting in soil saturation that leads to slope failure. The saturated soil becomes weak and loses cohesion, and the slope fails when rainfall crosses a certain threshold (Bernardie et al., 2014; Martinović et al., 2018; Lee et al., 2021). ~~This massive volume of soil causes enormous environmental degradation, infrastructure damage, and casualties, which is~~The heavy rainfall saturates a slope and triggers a ~~hindrance~~landslide due to ~~socio-economic aspect~~the reduction of the soil's shear strength and the increase of ~~the community (Van~~pore water pressure (Luino et al.,2022; Chen et al., 2021; ~~Alcántara-Ayala, 2021~~Chatra et al., 2019; Lacerda et al., 2014; Tsai and Chen, 2010). For example, steep slopes with loose soils and even moderate rainfall can lead to the displacement of an enormous quantity of soil mass. On the contrary, in slopes with more stable, cohesive soils, the surface failure might be smaller (Tsai and Chen, 2010). The rainfall quantity and duration influence the volume of the landslides; the higher the intensity and the longer the duration of rainfall, the larger the resulting ~~volume of landslides~~surface failure (Chen et al., 2017; Bernardie et al., 2014; Chang and Chiang, 2009). The landslide ~~occurrence~~occurrences can also be influenced by human activities that ~~fragilize~~weaken the slope, such as excavation at the slope toe and loading caused by construction and land use such as agriculture, mining etc. (Rosi et al., 2016). ~~Therefore,~~The rapid urbanization activities affect the topography through hill cutting, deforestation and water drainage (Rahman et al., 2017); these activities disturb the slope structure and change the water flow, which exacerbates the effect of landslides in regions where human engineering activities are mostly located (Holcombe et al., 2016; Islam et al., 2017; Chen et al., 2019). ~~accurate prediction of the~~

To estimate the volume of the soil mass displaceable subsequent to intensive rainfall, is essential to set appropriate mitigation strategies to reduce environmental degradation, infrastructure damage, casualties, and to establish post-disaster resilience policies to restore the socio-economic aspect of communities (Van et al., 2021; Alcántara-Ayala, 2021). This quantification of the volume of landslides due to rainfall (VLDR) is essential for effective risk

management (Tacconi Stefanelli et al., 2020), emergency response, engineering design (Cheung, 2021), economic assessment and environmental protection (Alcántara-Ayala and Sassa, 2023). Firstly, to manage landslide risk effectively, the quantification of VLDR can be useful for updating hazard maps to reflect the scale of potential landslides in various regions to facilitate the identification of high-risk zones for monitoring and intervention. In addition, to develop mitigation strategies, such as land stabilization measures and land use planning, planners might put in place strict construction regulations in particular regions that are susceptible to landslides (Mateos et al., 2020). The accurate measurements of VLDR can be used to promote public awareness for safety measures and preparedness (Yang and Adler, 2008). Secondly, estimating precise VLDR is crucial for structural engineers to design a structure that can withstand extreme landslide events. Knowing the exact volume of displaceable material, an engineer can set robust stabilization solutions to prevent future occurrences (Dai and Lee, 2001). Moreover, the VLDR can help design the drainage system to manage water flow by controlling groundwater and surface runoff to mitigate landslide risks (Dikshit et al., 2019; Kim et al., 2014). Furthermore, to prepare for emergence responses such as resource allocation, evacuation planning, and search and rescue operations, accurate VLDR estimation is necessary to ensure efficient implementation (Fan et al., 2019). To allocate resources effectively, the volume data is needed to determine the expected number of personnel for evacuation, materials sufficient for cleaning up and recovery (Amatya, 2016; Yang and Adler, 2008; Spiker and Gori, 2003). Further, to establish environmental protection measures such as ecosystem impacts, preservation of soil and water quality, and habitat restoration, the estimates of VLDR are essential (Pradhan et al., 2022; Li et al., 2022a; Barik et al., 2017).

To mitigate the economic impacts of landslides, the values of VLDR can be a basis for estimation of property damages, which is critical for settling insurance claims and assessment of financial impacts on communities and government to facilitate efficient budgeting for repairing damaged infrastructure and restoration of affected parts (Klimeš et al., 2017; Dai et al., 2002). The prediction of the VLDR can assist in long-term economic planning for landslide risk by creating disaster preparedness and recovery funds (Winter and Bromhead, 2012). The accurate estimation of the VLDR is an important key for designing strategies for resilience and planning for the protection of the inhabitants of a particular region with certain landslide risks subjected to a predicted quantity of rainfall (Conte et al., 2022). Consequently, for the safety of communities, the efficient selection of infrastructure construction sites must be done in places where landslides

~~cannot bury buildings~~with low landslide risks, (Fan et al., 2017). Further, for the protection of
crops, the farmland location, and other land use activities, accurate landslide prediction taking into
account real root causes through the analysis of triggering and influencing factors, is crucial to
achieve a durable landslide safety management system (Paudel et al., 2003; Lee, 2009; Fan et al.,
2017; Chen et al.,2019; Dai et al., 2019; Alcántara-Ayala, 2021).
~~The prediction of landslide volume due to rainfall is important for the analysis of~~
~~infrastructure placement to protect against being buried in extreme landslide events. In South~~
~~Korea, many infrastructures are placed at the foot of mountains, which makes them vulnerable to~~
~~extreme landslides, which can bury villages, farm lands etc. The findings of Lee (2016) indicated~~
~~that due to climate change, the average rainfall has increased by 271.23 mm for the period 1971-~~
~~2100 based on future climate scenarios. Therefore, the efficient prediction of landslide volumes~~
~~can be useful for land use management in such a way that locations with expected high volume of~~
~~landslide may be used for other activities which do not get affected by landslide events, such as~~
~~forest and gardens or activities that reduce water infiltration and non-continuous disturbance of~~
~~subsoil to maintain groundwater stability and strengthen the topsoil.~~
~~Most researchers focused on the prediction of landslides runout and susceptibility (Giarola~~
~~et al., 2024; Melo et al., 2019; Peruzzetto et al., 2020). Nevertheless, few researchers estimated the~~
~~volume of landslides based on the statistical approach (Ju et al., 2023; Dai and Lee, 2001). Ju et~~
~~al. (2023) constructed an area-volume power law model for the estimation of the volume of~~
~~landslides using LiDAR data in Hong Kong. Razakova et al. (2020) calculated landslide volume~~
~~using a digital elevation model and ground-based measurement. Dai and Lee (2001) found that the~~
~~12 hours of rainfall influenced the volume of landslides and frequency-volume followed the power~~
~~law relation. It was observed that most of these studies did not consider detailed predisposing~~
~~factors and their contribution to the prediction of the volume of landslides due to rainfall. Recently,~~
~~Lee et al. (2021) applied an artificial neural network (ANN) model for the prediction of the volume~~
~~of debris flow in the central region of South Korea based on the patterns from the already occurred~~
~~landslide characteristics and the region morphometry.~~ The prediction of VLDR has gained the
interest of many researchers to understand the mechanism and interaction between triggering and
aggravating factors. Saito et al. (2014) studied the relationship between rainfall-triggered
landslides to test whether the volume of landslides across Japan that occurred between 2001 and
2011 can be directly predicted from rainfall metrics. The findings revealed that larger landslides

occurred when rainfall exceeded certain thresholds, but there were significant discrepancies between peaks of rainfall metrics and maximum landslide volumes, and the total rainfall was the suitable predictor of landslides. Dai and Lee (2001) established the frequency-volume relation for landslides in Hong Kong and noticed that the relation for shallow landslides above 4m$^3$ followed the power law. The 12-hour rolling rainfall contributed most to the prediction of the volume of landslides. Ju et al. (2023) constructed an area-volume power law model for the estimation of the volume of landslides using high-resolution LiDAR data collected between 2010 and 2020 in Hong Kong. The aim was to estimate accurately the volume of landslides on small-scale landslides. The reliance on localized datasets limits the model's applicability in regions with different geological settings, and the model does not consider all variabilities of landslide characteristics. Razakova et al. (2020) calculated landslide volume using remote sensed data with the aim of assessing the efficiency of aerial photographs in environmental impact assessment and ground-based measurement. The study did not take into account the effect of vegetation and topography and only focused on a single landslide case, which may be a source of bias due to differences in soil composition and environmental factors. Hovius et al. (1997) analyzed multiple sets of aerial photos and frequency-magnitude relations for landslides in New Zealand. The finding pinpointed that the landslides frequency-magnitude followed power law and infrequent large magnitude contributed to the landscape change. The study also noticed the importance of soil composition in the size of the landslides. This work had a limitation due to the reliance on aerial photos only, which cannot provide accurate measurement in regions of dense forest, and the climatic conditions, which are landslide triggering factors, were not considered, and this may affect the generality of the findings. Guzzetti et al. (2008) applied statistical methods on regional landslide inventories and antecedent rainfall data ranging between 10 min to 35 days. The findings revealed that the slope angle and soil type significantly influence landslide volume estimates, and the rainfall intensity is more important than duration. Chatra et al., 2019) applied numerical methods to study the effect of rainfall duration and intensity on the generation of pore pressure in the soil; the finding revealed a higher instability in loose soil compared to medium soil slopes. The work only treated the interaction of soil and rainfall without considering the environmental factors and human activity, which might also influence mass failure. Recently, the application of GIS technologies has been increasing in the identification of regions susceptible to landslides (landslide zonation) (Chen et al., 2021; Gutierrez-Martin, 2020; Li et al., 2022b). These methods are essential in emergency

management because they provide a general overview of zones with a higher probability of landslide occurrence; however, they do not put emphasis on the determination of the approximate value of the volume of failing mass in relation to excessive rainfall events.

In the present study, the volume of landslides due to rainfall is predicted using OLS, RF, SVM, EGB, GLM, DT, DNN, KNN and RR algorithms, considering the details of triggering factors (i.e., rainfall) and predisposing factors (i.e., ~~geological,~~ geomorphological, soil and environmental).

~~In this study~~ Here, we aim to construct a data-driven algorithm that combines input parameters for physical-based and empirical models and ~~incorporate~~incorporates more complex non-linear features of input variables to predict the occurrence of associated events more accurately. The main assumption behind the data-driven algorithm is that the considered feature input of the model produces a similar volume of landslides due to rainfall and follows the same pattern at a particular region with the same features under the same quantity of rainfall. Here, we examine different machine learning algorithms and compare their performance using the coefficient of determinations $R^2$ ~~and mean square errors (MAE) resulting from the application of each algorithm. The model can be customized to be applied in other regions according to the regional settings~~$(R^2)$ and mean square errors (MAE), Root mean square error (RMSE), Mean absolute percentage error (MAPE) and symmetric mean absolute percentage errors of the predicted volume of landslides. The focus is to optimize the predictions of the volume of landslides due to rainfall, taking into account triggering and influencing factors with higher accuracy.

## 2. Data and Study ~~area~~Region

### *2.1. Study Region*

The region for testing the model is South Korea, characterized by mountainous (63% of total land) relief, especially in the eastern part of the country (Lee et al., ~~2021). The~~ 2022). South Korea is located on the southern part of the Korean ~~peninsula~~Peninsula, bordered by the Yellow Sea to the west coast and the East Sea (Sea of Japan) to the East. According to the Korean Meteorological Administration (2020), the country has a temperate climate ~~comprises cold and dry winters~~characterized by four distinct seasons: hot and humid summers~~.~~, cold winters, and springs and falls with moderate temperatures. The annual rainfall ranges between 1000 mm to 1400mm and 1800mm for the central region and southern region, respectively (Jung et al., 2017; Alcantara

and Ahn, 2020), During the summer ~~season~~, heavy rainfall from June to September leads to significant surface runoff, increases landslide risk, and causes approximately 95% of all landslides ~~due to rainfall~~ each year (Lee et al., 2020; Park and Lee, 2021). In addition, the landslides may be aggravated by typhoons, which mostly occur in August and September, and it is anticipated that frequency will increase due to climate change. (Kim and Park, 2021), The ~~annual~~ rainfall ~~ranges between 1000 mm~~trend analysis from 1971 to ~~1400mm and 1800mm for~~2100 predicted the ~~central region and southern region, respectively (Jung et al., 2017; Alcantara and Ahn, 2020). The geology~~increase in rainfall of 271.23mm, which indicates the ~~Korean peninsula is~~growing risk of landslides associated with climate change (Lee, 2016). Temperature variations are influenced by its geographical location, the average summer temperatures range between 25 and 30°C, while winter temperatures can drop to -10°C in some parts of the country (Korea Meteorological Administration, 2020). The South Korean geologically is mainly composed of granitic and metamorphic ~~(45%), igneous (30%)~~rocks, such as gneiss, schist, and ~~25% of sedimentary rocks (Lee and Winter,~~granite, which influence the stability of the landscape (Jung et al., 2024). The geomorphology is characterized by rugged mountains, river valleys, and coastal plains, with the Taebaek Mountains running along the eastern edge (Kim et al., 2020). In addition ~~2019). Subsequently~~, the influence of rainfall, environmental, geomorphology, and geological factors ~~frequently generated~~increase the vulnerability to landslides across the country, especially in the northeastern mountainous region, as depicted in Figure ~~1. The distribution of rainfall and volume is summarized in Fig~~ 1.

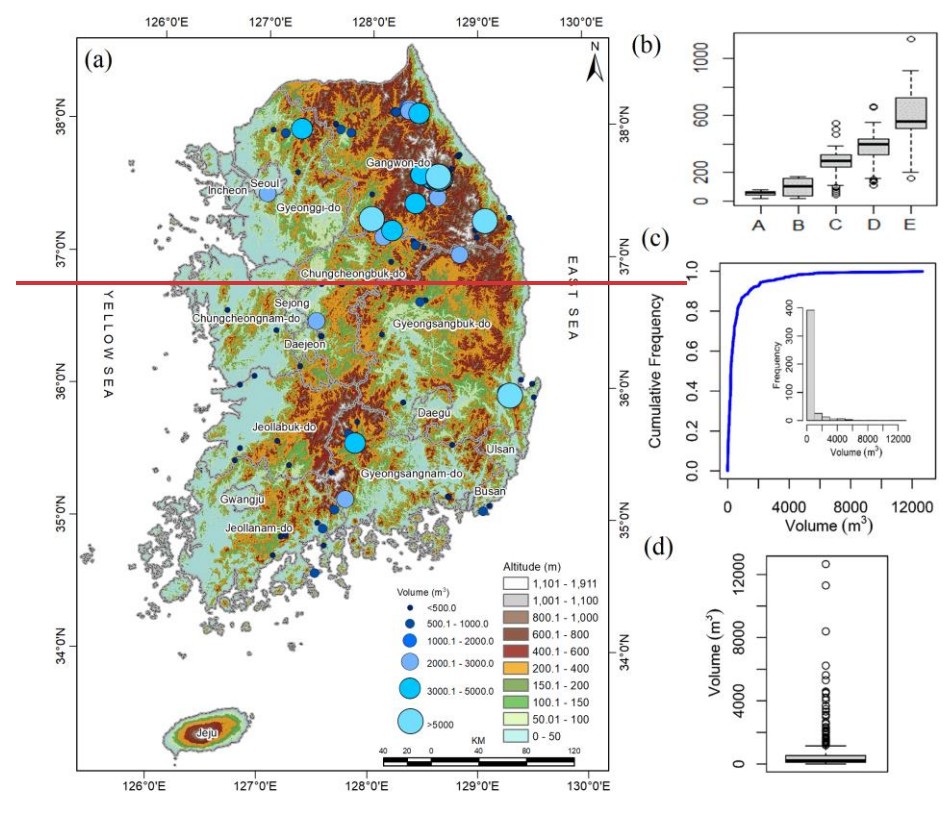

205       The predominant soil types in South Korea include clay, sandy, and loamy soils, each with

different characteristics affecting water infiltration, retention and erosion (Kang et al., 2022; Lee
et al., 2023). Clay soils, being more stable, can become highly saturated, increasing landslide risk
during heavy rains. On the other hand, sandy soils are more prone to shallow landslides due to fast
saturation, leading to instability. Regions with steep topography and poorly consolidated soil
(loose) are mostly at risk, especially after prolonged rainfalls (Kim et al., 2015).

211       Coastal areas are exposed to sea-level rise and coastal erosion, which can further

complicate the landscape and increase landslide susceptibility. The combination of heavy summer
rainfall, geological composition, and geomorphological factors makes South Korea particularly
vulnerable to shallow landslides. Thus, continuous monitoring and research are vital to understand
the complex interactions between climate, geology, soil types, and landslide occurrences in this
region (Park, 2022). Understanding the combination of environmental, geological stability, and
geomorphological features is crucial for developing effective disaster management strategies and
enhancing public safety in landslide-prone areas. As climate change continues to impact rainfall
patterns, South Korea faces ongoing challenges in mitigating landslide risks and protecting
vulnerable communities.

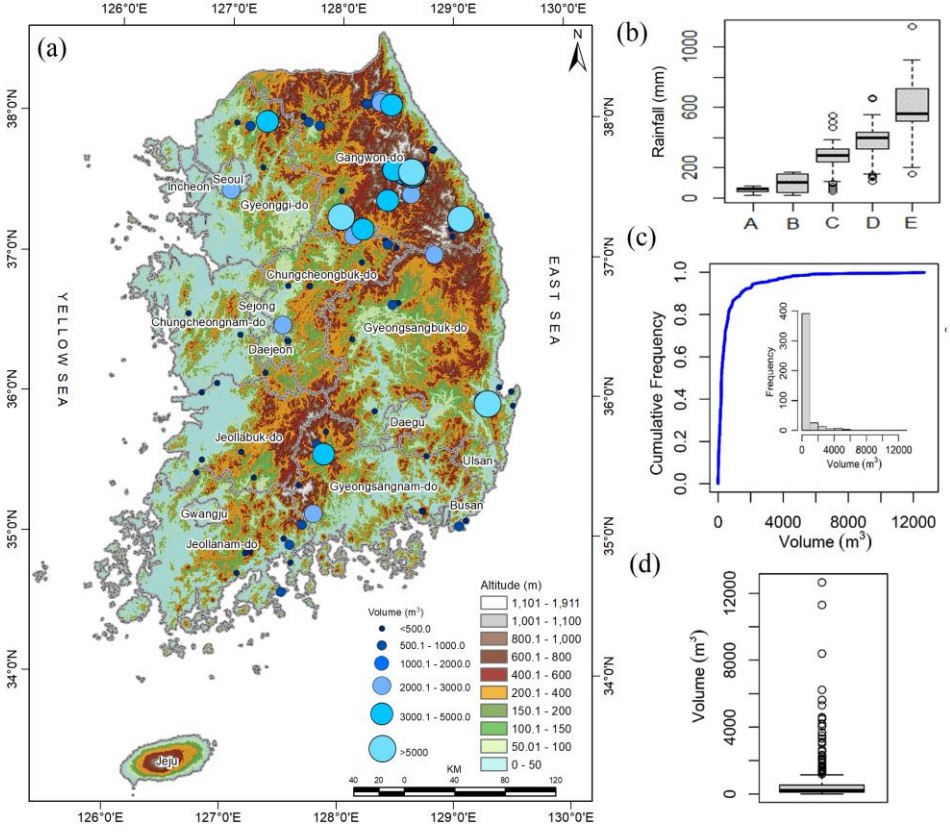

Figure 1. (a) Spatial distribution of landslides in South Korea, (b) temporal variation of rainfall, i.e., A: Maximum hourly rainfall, B: Four weeks rainfall, C: Three hours rainfall, D: Three days rainfall and E: Two weeks rainfall, (c) cumulative frequency distribution of volume of landslides and (d) box plot of volume of landslides.

### 3.2.2 *Data and method*

In this paper, we consider nine data-driven models, namely ordinary least square or Linear regression (OLS), random forest (RF), support vector machine (SVM), extreme gradient boosting (EGB), generalized linear model (GLM), decision tree (DT), and deep neural network (DNN), k-nearest neighbor (KNN) and Ridge regression (RR) to predict the volume of landslides due to rainfall. The model is tested on the South Korean landslides inventories and predisposing factors coupled with triggering factors, i.e., rainfall data. The detailed workflow is summarized in Figure 2. The steps for construction of these models can be briefly summarized as follows: a) the dataset for landslide inventories is cleaned and joined with rainfall dataset, b) the collinearity analysis is made using variance inflation factor, c) continuous feature are scaled (Z-score) (Bonamutial and Prasetyo, 2023) to facilitate algorithms to converge fast, d) the dataset is split into training and test set, e) all models are tested on the same training set, and the model evaluation on the test set using MAE and $R^2$ for the comparison of actual and predicted volume by each model, f) variable importance is calculated for most performing model, and g) the distance correlation is calculated for each continuous feature, and Kruskal-Wallis and Dunn test are conducted to examine the similarity of the effect of each category on the landslide volume.

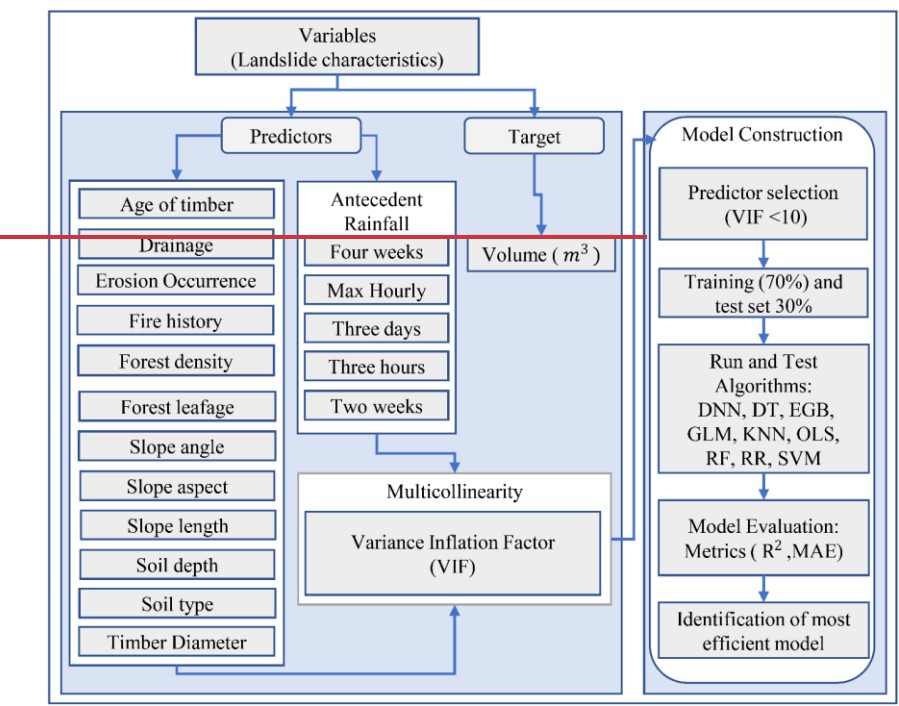


~~Figure 2.Workflow for the prediction of volume of landslide due to rainfall.~~

*3.1 Data*
The landslide inventory dataset contains ~~450~~455 landslide record information from 2011 to 2012,
~~which was~~ collected from different locations in South Korea by Korean Forest Services. This
dataset tabulates information on landslide ~~location, volume, slope~~geometry, such as runout length,
~~soil type, drainage situation, fire history, and~~width, depth, and volume of the affected area, along
with geomorphological composition, vegetation ~~features such as age, diameter of timber, leafage,~~
~~and forest density.~~, and antecedent rainfall prior to landslide events. The details regarding landslide
predisposing and triggering factors are summarized in Table 1.
The majority of landslides in this region were shallow, translational slope failures (Kim
and Chae, 2009; Kim et al., 2001). The occurred landslides had a volume varying between 1.5m$^3$
to 12,663m$^3$ and predominantly occurred in the northeastern and southeastern region (Fig.1a,c &
d). The ~~outcome variable (volume) to be predicted was estimated as a product of the area affected~~

occurred landslides were hallowed and skewed to the right with 2570.7m$^3$ as 95$^{th}$ quantile, largest volume was 12,663m$^3$, and the aggregate mass of landslide due to rainfall was 276,986.62m$^3$. The estimation of the volume of removed material by landslides is important as it helps to assess risks the estimated damage can cause down at the toe of the failed slope, such as blocking transportation network, burying crops or farmland, the damage-built environment near landslide risks area, and post-disaster recovery planning (Evans et al., 2007; Rotaru et al., 2007; Intrieri et al., 2019).

Table 1. Landslide influencing and triggering factors.

| Group | Features | Description | Reference |
|---|---|---|---|
| Vegetation | Fire history | The burning of the vegetation intensifies the mass movement of soil near the uncovered burned stem of trees and free movement on uncovered soil due to post-fire rainfall and storms. The sliding may also be due to loss of vegetation, altered soil property and structure, which lead to soil degradation and infiltration which increase pore pressure, and change in hydrology by concentrating water flow in places that exacerbate landslides. | Highland and Bobrowsky, 2008; Culler et al., 2021; Hyde et al., 2016; Stoof et al., 2012 |
| | Age of tree | Mature forests have more resistance to shallow landslides due to highly developed roots, which improve soil cohesion and leaves that prevent direct contact of raindrops with the soil surface. | Sato et al., 2023; Lann et al., 2024 |
| | Forest density | The presence of forest reduces the likelihood of landslides about three times compared to grassland. Grassland has been revealed to be three times more vulnerable to shallow landslides than broadleaf and, coniferous and in secondary forests. | Lann et al., 2024; Greenwood et al., 2004; Turner et al., 2010; Scheidl et al., 2020; Asada et al., 2023 |
| | Timber diameter (m) | Tree spacing and size had been used to investigate the effect of root and tree in shallow landslide control. The high root density generally enhances slope stability, | Cohen and Schwarz., 2017; Wang et al., 2016 |

| | | and specific tree placement and root sizes between 5 to 20 mm are effective in landslide prevention. | |
|---|---|---|---|
| | Drainage | The drainage has a significant effect on the slope stability and promotes the efficient control of the influence of rainfall on the ground water fluctuation. The presence of drainage increases the threshold of landslides due to rainfall. | Yan et al., 2019; Sun et al., 2010 ; Wei et al., 2019 ; Korup et al., 2007 |
| Geomorphology | Slope angle (degree) | The steeper slopes have lower presence of landslide due to low transportable materials. Slopes between 20-40 degrees are most vulnerable to greater landslides as rainfall intensity and duration increase. Here, we considered the average angle of the terrain at the landslide location, which provides valuable insight into the region's overall steepness and geomorphic characteristics, which are crucial factors influencing landslide susceptibility and risk modeling. | Duc, 2013 ; Qiu et al., 2016 ; Donnarumma et al., 2013 |
| | Slope aspect | The effect of rainfall on slope differs by slope angle and slope aspect which lead to unevenly distributed occurrence of landslides. | Panday and Dong, 2021; Cellek, 2021 |
| | Slope length (m) | The volume increases as the slope length increases. There exists a complex interplay between rainfall, length of slope and slope angle on the occurrence of landslides. | Turner et al., 2010 |
| | Soil depth (m) | Soil properties, depth, and texture have significant differences in infiltration rates, which have different influences on the occurrence of landslides. | Kitutu et al., 2009; McKenna et al., 2012 |
| | Soil type | Higher rainfall intensity affects the occurrence of landslides differently, particularly in certain soil types that have shorter saturation and failure times. | Liu et al., 2021 |
| Location | Altitude | Regional variability of elevation and mountain steepness affect the quantity of rainfall and associated landslides. | Hyun et al, 2010, Yoon and Bae, 2013; Park, 2015 Um et al., 2010 |

| | | | |
|---|---|---|---|
| Rainfall | Maximum hourly rainfall | The rainfall infiltrates the slope and increases pore water pressure that reduces soil shear strength, which leads to soil saturation that causes surface failure. | Wieczorek, 1987; Smith et al., 2023; Dai and Lee, 2001; Smith et al., 2023 |
| | Continuous rainfall | Sudden intense rainfall concentrated in short periods of time is responsible for shallow landslide and debris flow. | Zhang et al., 2019 |
| | Three hours rainfall | | |
| | Three days rainfall | The antecedent rainfalls increase moisture in the soil and weaken soil cohesion. | Ran et al., 2022 Zhang et al., 2019; Bernardie et al., 2014; Chen et al., 2015a; Gariano et al., 2017 |
| | Two weeks rainfall | | |
| | Four weeks rainfall | | |

Location parameters such as altitude, latitude and longitude are essential elements that determine the microclimate of a given region, influencing rainfall patterns (Hyun et al., 2010; Yoon and Bae, 2013; Park, 2015). The northeastern region is characterized by high-elevation terrain, such as Taebaek, and Sobaek ranges, which dry air and lead to orographic precipitation (Yun et al., 2009). The windward mountain versants receive a substantial amount of rainfall, which can increase the likelihood of landslides (Jin et al., 2022). This variation of rainfall with respect to the direction highlights the importance of including slope aspect variables in landslide studies (Kunz and Kottmeier, 2006). Figure 2(g) depicts the relationship between the slope aspect and the volume of landslides and slope aspect, altitude and fire history and shows that larger volumes were localized in regions that faced forest fire and altitudes between 500 and 1000m. Additionally, the topographical features such as slope length and slope angle affect the size of the landslide (Panday and Dong, 2021), slope failure due to over-saturation from groundwater and rainfall infiltration that destabilize the slope (Kafle et al., 2022). Furthermore, slope length, slope angle and slope aspect play an important role in the determination of the volume of geological material uprooted by landslides (Zaruba and Mencl, 2014; Khan et al., 2021). The slope stability depends on thesoil composition properties of composing material which have different, including soil permeability index which indicatesindices that affect water infiltration capabilityand saturation level (Chen et al., 20152015a). From surveyed regions, three main soil types, namely, sandy loam, loam, and silt loam, were observed, and their coefficient of permeability is 1.7, 1.65 and 1.5, respectively (Lee

et al., 2013~~), were used as numerical predictor variables. In addition,~~). Moreover, to reduce the infiltration drainage network that channeling rainwater ~~in hilly~~ terrain drains soil and reduces the saturation, which minimizes the likelihood of landslide occurrence as a result of groundwater discharge and rainfall water flow (Hovius et al.,~~1998~~ 1997; Wei et al., 2019). Furthermore, the ~~occurrence of forest fires can contribute to the occurrence of landslides due to the burning of~~ vegetation ~~covering the area and can also change soil property and increase soil pH (Lee et al., 2013). Moreover, the vegetation type, leafage, roots, age and density can be predictors of the occurrence and the volume of landslides. The vegetation covers~~protects the topsoil, ~~prevents drying and~~ from the direct ~~hit of rain drops which automatically dig holes on~~impact of raindrops hitting the ground, which causes erosion due to the force of gravity ~~acting on the raindrop combined with the soil permeability~~and reduces infiltration (Omwega, 1989; Keefer, 2000). The absence of vegetation allows rainwater to seep away fine topsoil, causing shallow landslides (Gonzalez-Ollauri and Mickovski, 2017). ~~Thus, planting~~On the contrary, vegetation ~~is recommended as a better practice to improve~~improves soil cohesion and ~~prevent~~prevents potential shallow landslides due to soil-~~-~~root interaction (Gong et al., ~~2017~~2021; Phillips et al., 2021). The density of vegetation (forest) and leafage type (broad, pines or mixture) ~~determine~~directly affects the quantity of raindrop intercepted and prevented from directly hitting ~~directly~~ the soil, which emphasizes the vegetation's landslides mitigation role. ~~The rainfall, a triggering factor~~Further, the occurrence of forest fires can contribute to the occurrence of landslides ~~which consists of rainfall at~~due to the ~~time~~burning of ~~landslide event~~vegetation covering the area, changing soil properties, and ~~antecedent rainfall are critical factors that influence the occurrence of landslides (Yune~~increasing soil pH (Lee et al., 2013)~~2010; Khan et al.,2012; Kim et al., 2021). In this study, we consider time-based aggregated rainfall. The considered variables are illustrated in Table 1.~~

The rainfall, a triggering factor of landslides, is the immediate cause of slope instability and failure due to infiltration that leads to saturation resulting from increased pore water pressure that reduces soil shear strength (Yune et al., 2010; Khan et al., 2012; Kim et al., 2021; Lee et al., 2021). The antecedent rainfall increases the moisture in the soil, which accelerates the soil saturation; the cumulative effect is essential to understand the saturation levels (Ran et al.,2022). In this study, rainfall variables are grouped based on time, namely, continuous rainfall, which is the accumulative value of rainfall on the day of a landslide from rainfall start hour to the landslide event, maximum hourly rainfall, rainfall during the fixed period such as three hours, one day, three

days, two weeks etc (Fig. 1b). The histograms for rainfall considered in this study are depicted in
Figure 2(a-f), and the descriptive statistics for all continuous variables are in Table 2.

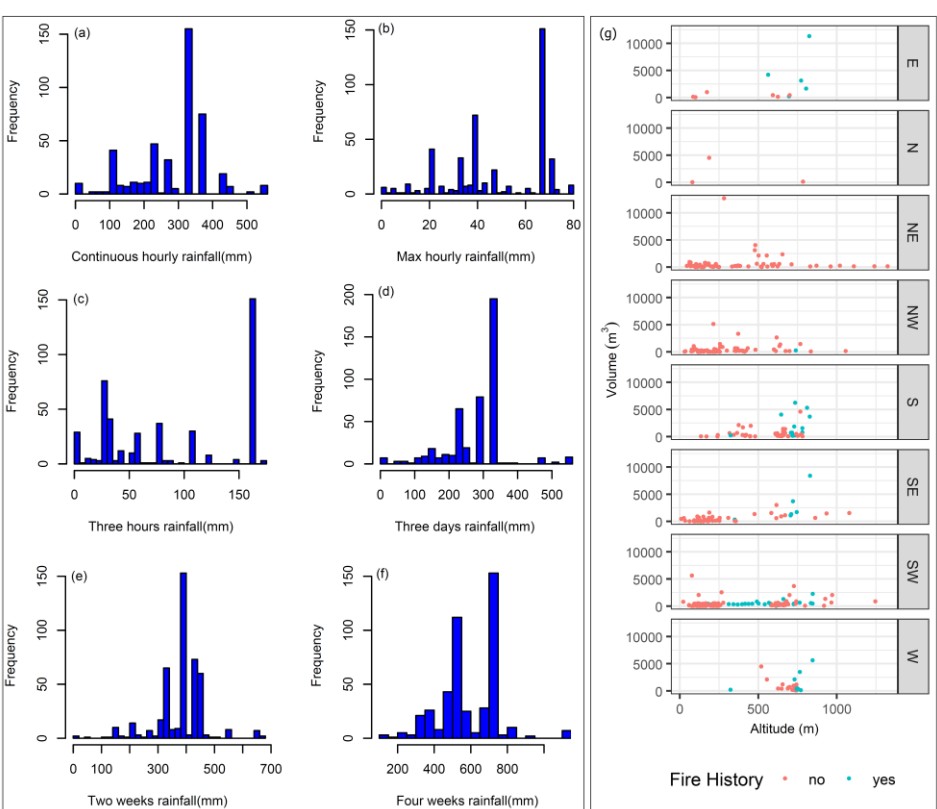

Figure 2. (a-f) Histograms of rainfall data, and (g) the scatter plot showing the variation of landslide
325            volumes with respect to slope aspect, fire history and altitude.


Table 1. Considered variables for data-driven model construction.

| Group | Features | Description | Reference |
|---|---|---|---|
| Vegetation | Fire history | The burning of the vegetation intensifies the mass movement of soil near uncovered burned stem of trees and free movement on uncovered soil due to post-fire rainfall and storm. | (Highland and Bobrowsky, 2008; Culler et al.,2021) |
| | Age of tree Forest leafage Forest density Timber diameter (m) | The age of tree combined with the quantity of rainfall may generate higher landslide intensity especially in in trees of age below 10 years. The disturbance of vegetation significantly impacts the susceptibility of landslides in forested regions. | Turner et al., 2010 ; Scheidl et al.,2020 |
| Geomorphology | Drainage | The drainage has a significant effect on the slope stability and promote the efficient control of the influence of rainfall on the ground water fluctuation. The presence of drainage increases the threshold of landslides due to rainfall. | Yan et al., 2019 ; Sun et al.,2010 ; wei et al.,2019 |
| | Erosion | The presence of erosion increases and contributes to the destructive capability of landslides by increasing the volume of transported materials. | Korup et al., 2007 |
| | Slope angle (degree) | There exists an established relationship between the slope morphology and volume | |

| ~~Group~~ | ~~Features~~ | ~~Description~~ | ~~Reference~~ |
|---|---|---|---|
| | ~~Slope aspect~~ ~~Slope length (m)~~ | ~~of landslide due to rainfall. The volume increases as the slope length increases. the steeper slopes have lower presence of landslide due to low transportable materials~~ | ~~Qiu et al.,2016 ; Donnarumma et al., 2013~~ |
| | ~~Soil depth (m)~~ ~~Soil type~~ | ~~Soil properties, depth and texture have a significant difference in infiltration rates which generate different influence on the occurrence of landslides.~~ | ~~Kitutu et al., 2009 ; McKenna et al., 2012~~ |
| ~~Rainfall~~ | ~~Maximum rain~~ ~~Four weeks rain~~ ~~Three hours rain~~ ~~Three days rain~~ ~~Two weeks rain~~ | ~~Rainfall intensity has an effect on the volume and frequency of landslides being the major triggering factor. The antecedent rainfall and duration of rainfall influence the volume, and deep landslides happen due to rainfall of long duration.~~ | ~~Wieczorek, 1987; Dai and Lee, 2001; Bernardie et al., 2014; Gariano et al.2017~~ |


~~Variable selection procedure was carried out based on previous literature and applied in the~~
~~model using variance inflation factor (VIF) (O'Brien, 2007) to eliminate collinear variables. The~~
~~variable with VIF<10 was considered as non-colinear and hence used in the model. The summary~~
~~statistics of variables with VIF <10 were summarized in Table 2. The training and test set was~~
~~scaled (Z-score or variance stability scaling) to solve convergence issues that are associated with~~
~~running the model without feature scaling (Singh and Singh, 2022). To run the model on the data~~
~~data driven methods that accept numerical features, the test and training set was one-hot encoded~~
~~to create a feature matrix (Seger, 2018).~~

Table 2: Summary statistics continuous variables.

| Variable | units | N | Min | Mean | Median | Max | Std dev |
|---|---|---|---|---|---|---|---|
| Max Hourly rain | mm | 455 | 18.50 | 48 | 58 | 78 | 20 |
| Continuous rainfall | mm | 455 | 0 | 285 | 327 | 550 | 106 |
| Three hours rainfall | mm | 455 | 15 | 88 | 100 | 171 | 60 |
| Twelve Hours rainfall | mm | 455 | 0 | 150 | 99 | 447 | 95 |
| One day rainfall | mm | 455 | 0 | 202 | 162 | 538 | 112 |
| Three days rain | mm | 455 | 0 | 280 | 284 | 550 | 86 |
| Seven days rain | mm | 455 | 4 | 332 | 330 | 534 | 88 |
| Two weeks rain | mm | 455 | 0 | 388 | 400 | 663 | 90 |
| Three weeks rain | mm | 455 | 86 | 504 | 533 | 914 | 115 |
| Four weeks rain | mm | 455 | 158 | 587 | 561 | 1135 | 160 |
| Soil depth | m | 455 | 0.2 | 0.6 | 0.75 | 0.75 | 0.19 |
| Soil type | | 455 | 1.5 | 1.67 | 1.75 | 1.7 | 0.087 |
| Timber diameter | m | 455 | 0.15 | 0.27 | 0.23 | 0.35 | 0.086 |
| Age of tree | Years | 455 | 10 | 35.234 | 35 | 60 | 14 |

| Variable | units | N | Min | Mean | Median | Max | Std dev |
|---|---|---|---|---|---|---|---|
| ~~Volume~~Slope length | ~~m³~~m | ~~450~~455 | 1.~~5~~8~~5~~ | ~~599.59~~21 | ~~211.68~~13 | ~~126631~~80 | ~~1237.12~~8 23 |
| Slope angle | Degree (º) | 455 | 10 | 34 | 34 | 65 | 7.9 |
| Altitude | m | 455 | 9 | 391 | 272 | 1324 | 273 |

## 3.~~2 *Method*~~ **Methods**

In this paper, we consider nine data-driven models, namely OLS, RF, SVM, EGB, GLM, DT, DNN, KNN and RR to predict the volume of landslides due to rainfall. The model is tested on the South Korean landslides inventories and predisposing factors coupled with triggering factors, i.e., rainfall data. The detailed workflow is summarized in Figure ~~In this study, nine data-driven methods were selected and tested on a Korean dataset. This section contains a brief introduction to each tested method.~~ 3. The steps for construction of these models can be briefly summarized as follows: a) the dataset for landslide inventories is cleaned and combined with rainfall dataset, b) the collinearity analysis is made using variance inflation factor, c) continuous feature are scaled (Z-score) (Bonamutial and Prasetyo, 2023) to facilitate algorithms to converge fast, d) the dataset is split into training and test set, e) all models are tested on the same training set, and the model evaluation on the test set using mean absolute error (MAE), coefficient of determination ($R^2$), root mean square error (RMSE), symmetric mean absolute percentage error (SMAPE) and mean absolute percentage error (MAPE) for the comparison of actual and predicted volume by each model, f) variable importance is calculated for most performing model, and g) the distance correlation is calculated for each continuous feature, and Kruskal-Wallis and Dunn test are conducted to examine the similarity of the effect of each category on the landslide volume.

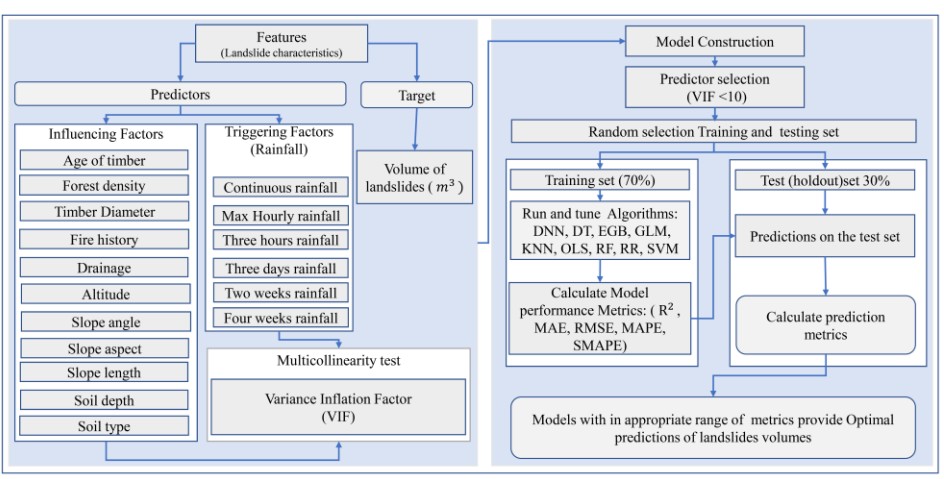

Figure 3. Workflow for the prediction of the volume of landslides due to rainfall.

### *3.1 Model Construction*

In the present investigation, we aimed at predicting the volume of landslides using models that minimize error with interpretability and scalability. Since one model can not have all properties at the same time, we decided to select some of the models with those properties. The OLS, GLM, and DT were widely used for their high interpretability, which helps to understand the influence of individual features on predictions (Gelman, 2007; Breiman, 2017). On the other hand, the EGB, RF, SVM, RR, and KNN were used due to their robust performance in capturing complex patterns in data, which is essential for accurate predictions of landslide volumes (Chen and Guestrin, 2016; Liaw and Wiener, 2002; Hastie, 2009). Additionally, considering that the model will be used on a regional scale, which will require big data, the EGB, RF, and DNN are designed to efficiently handle large datasets, making them suitable for the regional scale analysis. These last models can be scaled to incorporate more data from different geographical areas without significant adjustments, enhancing their applicability in future research (Krizhevsky et al., 2012). Accordingly, nine data-driven methods were selected and tested on a Korean dataset to predict VLDR.

The first considered method is OLS, which is applied to estimate parameters of multilinear regression that yield the minimum residual sum of squares errors from the data (Dismuke and Lindrooth, 2006) under assumptions of no correlation in independent variables and in error term,

constant variance in error terms, non-linear collinearity of predictors, and normal distribution of error terms. The RF-regression is a supervised data-driven technique based on ~~the~~ ensemble learning, which ~~construct~~constructs many decision trees during ~~the~~ training time of a model by combining multiple decision trees to produce an improved overall result of the model outcome. The RF-regression is more efficient in the analysis of multidimensional ~~dataset~~datasets (Borup et al., 2023). RF is an effective predictive model due to non-overfitting characteristics based on the law of large numbers (Breiman, 2001). The decision tree (DT) regression is a predictive modeling technique in ~~a~~the form of a flowchart-like tree structure ~~of~~that includes all possible results, output, predictor costs, and utility. The DT simplifies the decision-making due to its algorithm that mimic human brain decision-making patterns (Rathore and Kumar, 2016). The KNN technique draws an imaginary boundary in which prediction outcomes are allocated as the average of k-nearest point predictors and averaging their output variable (response). The KNN calculates Euclidian distances to identify likeness between datapoints and then it groups points that have smaller distances between them (Kramer and Kramer, 2013). The RR is an improved form of ordinary least square, which serves to respond to ~~the case~~cases where ~~the~~ collinearity is found in predictor variables. The estimated coefficients of ridge are biased estimators of true coefficients and are generated after adding a penalty on the OLS model. The RR has always lower variances compared to OLS (Saleh et al., 2019). The advantage of the GLM over OLS is that the dependent variable need not follow the normal distribution. The GLM is composed by random and systematic components, and the link function that links the two. In this study, the GLM with Gaussian link function was applied. GLM are fitted using maximum likelihood estimation (Dobson and Barnett, 2018). The DNN are among data-driven models that revolutionized different fields; the DNN learns via multi-processing layers and identifies intricate patterns in the data to predict the outcome (LeCun et al., 2015). Here, the backpropagation algorithm was used to predict the estimated outcome. The advantage of DNN is to discover the complex structures in the data using a back propagation algorithm with the capability to change the internal parameter (weight update). The SVM is popular for balanced predictive performance which makes it capable to train model on small sample size (Pisner and Schnyer, 2020). SVM has been applied in many different landslide studies (Pham et al., 2018; Miao et al., 2018). SVM methods identify the optimal hyperplane in ~~multi-dimensional~~multidimensional space that separates different groups in the output values. The EGB is the most powerful and leading supervised machine learning method in solving regression

problems. It can perform parallel processing on ~~windows~~Windows, and Linux (Chen et al., ~~2015~~2015b). The gradient boosting trains of differentiable loss function, and the model fits when the gradient is minimized. In this paper, both traditional statistical predictive models and machine learning models were used. The firsts are known for high clarity and ~~explain ability~~explainability, and the second is famous for handling non-linearity in features. In some cases, the performance of advanced data-driven algorithms is almost similar (Chowdhury, et al., 2023).

### *3.2 Feature selection and data splitting*

The variable selection procedure was carried out based on previous literature and applied in the model using generalized variance inflation factor (GVIF) (O'Brien, 2007) to eliminate collinear variables. The variable with GVIF<10 was considered non-colinear and used in the model. Figure 4 depicts retained features and corresponding GVIF values. The retained features have GVIF less than 10 (O'brien, 2007). Accordingly, all depicted variables were considered for the model training. Further, to train the model, the datasets were split randomly, with 70% of the data for the training set and 30% for testing (Nguyen et al., 2021). The 10-fold cross-validation was performed to obtain an optimal model.

The training and test set was scaled (Z-score or variance stability scaling) to solve convergence issues that are associated with running the model without feature scaling (Singh and Singh, 2022). To run the model on the data using driven methods that accept numerical features only, the test and training set was one-hot-encoded to create a feature matrix (Seger, 2018).

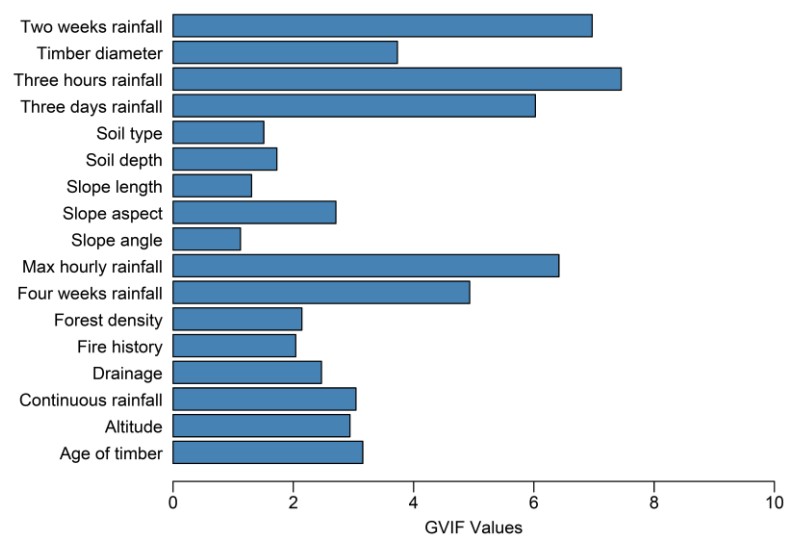


Figure 4. Generalized Variance Inflation Factor (GVIF) bar plot for features.
*3.3 Model evaluation metrics*

443   The model performance evaluation is a process of quantifying the difference between the

observed value not used in the modeling process and the predicted value by the model. Different
metrics are applied depending on the type of task, whether it is a classification or a regression
problem. Subsequently, the widely used evaluation metrics for regression models, namely, $R^2$,
MAE, RMSE, MAPE and SMAPE, were utilized to evaluate the model performances. The metric
formulae and evaluation criteria are summarized in Table 3.
**Table 3.** Model evaluation metrics.

| Metrics | Evaluation | Reference |
|---|---|---|
| $RMSE = \sqrt{\dfrac{1}{n}\sum_{i=1}^{n}(y_i - \hat{y}_i)^2}$ | • Measures the square root of the average squared differences between predicted and actual values.<br>• Lower values indicate better model performance. | Hyndman and Koehler, 2006. |
| $MAE = \dfrac{1}{n}\sum_{i=1}^{n}|y_i - \hat{y}_i|$ | • The average of the absolute differences between predicted and actual values.<br>• Lower values indicate better model performance. | Willmott and Matsuura, 2005 |

| | | |
|---|---|---|
| $$MAPE = \frac{100}{n}\sum_{i=1}^{n}\left|\frac{y_i - \hat{y}_i}{y_i}\right|$$ | • Measures the accuracy of a model as a percentage, which can be more interpretable.<br>• Lower values indicate better model performance. | Armstrong, 2001 |
| $$SMAPE = \frac{100}{n}\sum_{i=1}^{n}\frac{|y_i - \hat{y}_i|}{|y_i| - |\hat{y}_i|}$$ | • Unlike MAPE, which can be skewed by very small actual values, SMAPE accounts for both the actual and predicted values, making it symmetric.<br>• SMAPE is expressed as a percentage<br>• Mitigates the impact of small actual values on the error metric, providing a more balanced assessment.<br>• Lower values indicate better model performance. | Hyndman and Koehler, 2006 |
| $$R^2 = 1 - \frac{\sum_{i=1}^{n}(y_i - \hat{y}_i)^2}{\sum_{i=1}^{n}(y_i - \bar{y})^2}$$ | • Represents the proportion of variance in the dependent variable that can be explained by the independent variables.<br>• Values closer to 1 indicate a better fit | Darlington, 1990; Chicco et al., 2021 |

\*$y_i$ and $\hat{y}_i$ representing the actual and predicted value and, $\bar{y}$ and $n$ standing for the mean of actual value and number of observations in the dataset, respectively.

## 4. Results

Prior to the construction of the model, the collinearity analysis was performed and variable with less variance inflation factor were retained for training and testing models. Figure 3 depicts retained features and corresponding VIF values. The retained features have VIF less than 10 (O'brien, 2007). All predictors except three days rainfall exhibited VIF less than 5 and still less than 10. Accordingly, all depicted variables were considered for predictive model construction.

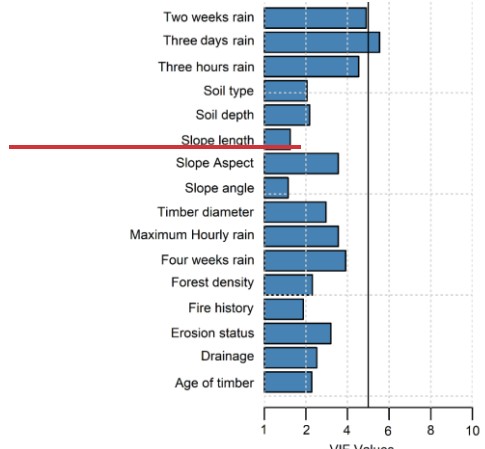


Figure 3. Variance inflation factor bar plot for explanatory variables.

461  The model was developed in R with different libraries, as discussed below. The DNN

regression model was constructed using dnn() function from the cito library (Amesoeder et al.,
2023), with threetwo hidden layers of (50,50, 50) nodes. ModelThe model was trained on
208L1500L epochs, learning rate (lr = 0.101), and loss = "mae". The decision tree regression model
was constructed with tree() function from the tree library, with the recursive-partition method. The
ridge regression model was constructed using glmnet() function from glmnet library(Jerome
(Friedman et al., 2010). theThe optimal lambda was obtained by performing 10-fold cross-
validation. The EGB model was built using xgboost() function in xgboost packagagespackage
(Chen et al., 2022). The optimal model was obtained at 357th524th boosting iteration with allmax
depth = 5 and other parameters set to default. The GLM regression model was constructed using
glm() function( (R core Team, 2022) with family gaussianGaussian and identitylog link to
constrain the model of predicting positive outcomes. The KNN regression was constructed using
knnreg() function from the caret package (Kuhn, 2022,), with number of neighbors (, k=7).17. The
OLS model was constructed lm() from the stats package (R core Team, 2022). The RF model was
run using randomForest() from the randomforest package (Liaw and Wiener, 2002,) with default
parameters and the optimal model was reached at 63rd256th iteration. The ridge regression model
was constructed using glmnet() from the glmnet package (JeromeFriedman et al., 20122010), with
ridge penalty (alpha=0). The SVM regression model with linear kernel was built using e1071
package (Meyer et al., 2021) and other parameters set to default.
The predictive performance of all tested models ~~was summarized in~~ on the holdout dataset
is depicted by the scatterplot (Fig. 4̶5) of actual volume as recorded in the test set and predicted
outcome values of each model. The red line represents the perfect prediction. The scatter plot of
actual and predicted values of tested models shows that OLS performed least compared to other
models with $R^2$=0.2̶7̶2744, that is, 2̶7̶29% of ~~variance~~variances in the model ~~could be~~were
explained by ~~predictor variables.~~predictors. The second least performing was ~~GLM~~the RR with
$R^2$= 0.2̶9̶ ̶t̶h̶a̶t̶3034, which is 2̶3̶.6% improvement compared to OLS. Among all models ~~five~~, three
out of nine, namely, OLS, ~~KNN, GLM,~~ SVM, and RR, performed below 50%; however, these
models predicted well small values of volume (below 2000m$^3$). The MAE of these ~~five~~three
models was higher than the remaining ~~four~~six models, namely DNN, DT, GLM, KNN, RF, ~~DNN~~
and EGB. Among these lasts, the most performing was EGB with $R^2$= 0.8̶5̶88 of variance explained
by predictors and MAE=2̶4̶5̶146.6 m$^3$. The ~~summary of coefficients of determination~~evaluation
metrics for the training and ~~mean absolute errors for~~ tested models are summarized in Table 3̶.̶ 4.
Considering the $R^2$, the three models, namely EGB, RF, and DNN, had a value of $R^2$ above 80%
on the holdout set.

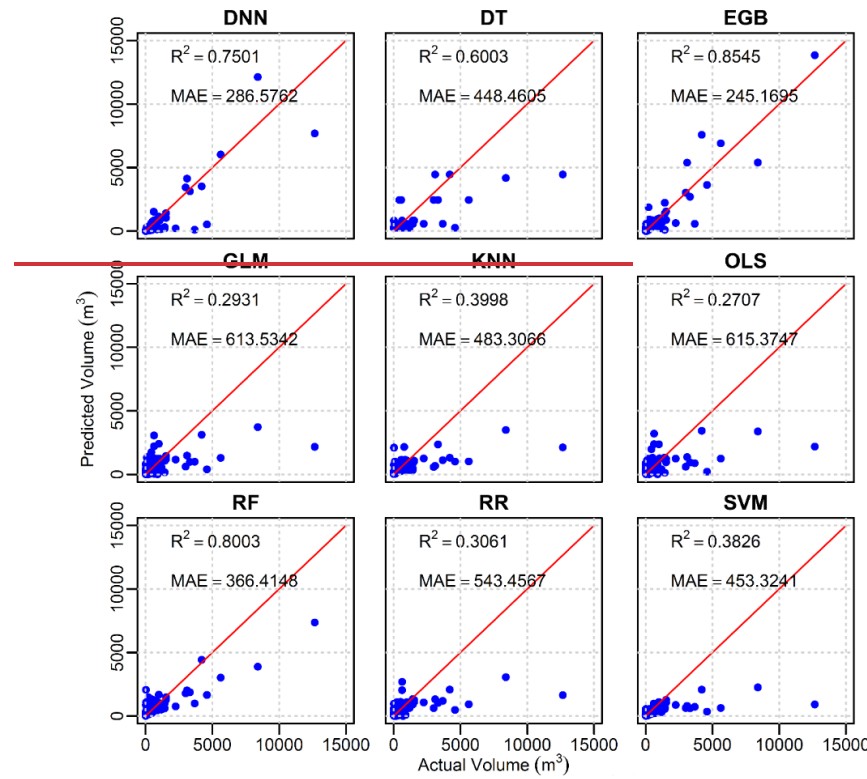


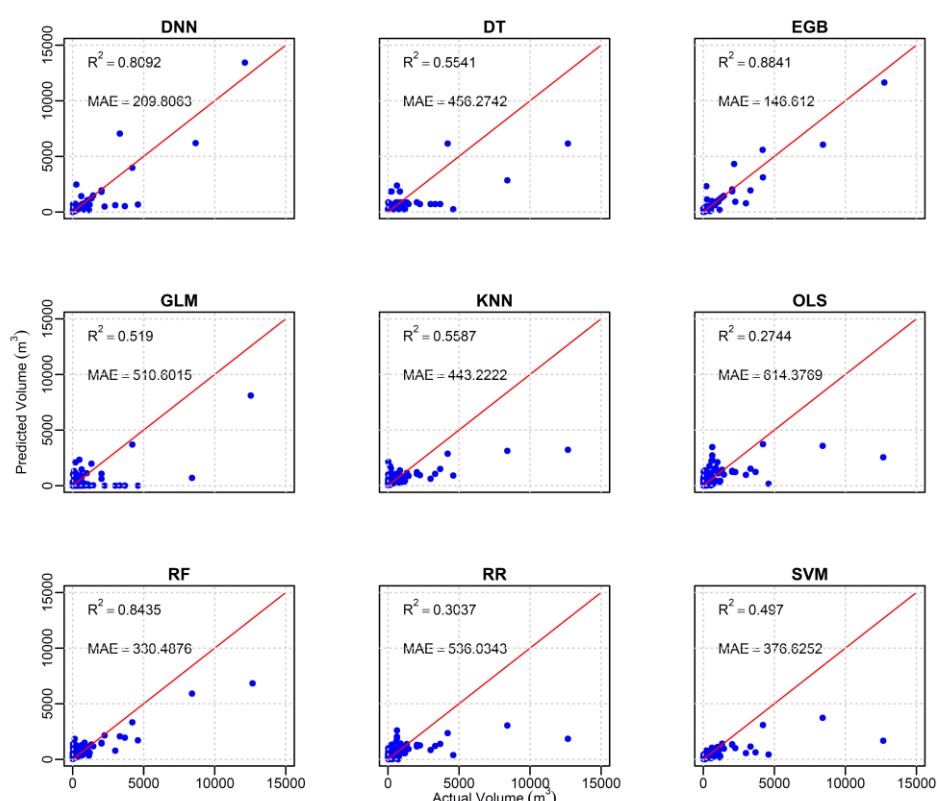


Figure 45. Scatterplot of actual and predicted values for nine tested models.



Regarding the prediction on the training set, the GLM had an $R^2$ of 83%. Nevertheless, the
prediction on the holdout set was 51.9%; this large variation in variance explained by predictors
indicates that the GLM model did not catch all non-linear patterns in the holdout set. It is
noteworthy that the prediction difference in $R^2$ on both training and test for the random forest
exhibited a very small difference compared to EGB and DNN, that is, 1.75% compared to 12.17%
and 7.72% for DNN and EGB, respectively. Despite the stable prediction of RF, the performance
in terms of SMAPE, the DNN was the second lowest symmetric mean absolute percentage error,
43.83m$^3$ and 39.79 m$^3$ on training and test sets, respectively. According to Chicco et al. (2021), the
$R^2$ is more informative in regression modeling; thus, RF had better predictions than the DNN.

Table ~~34~~. Summary of ~~R2 and MAE~~prediction metrics for tested models on the training and test
set.

| Metrics / ~~Models~~ | | DNN | DT | EGB | GLM | KNN | OLS | RF | RR | SVM |
|---|---|---|---|---|---|---|---|---|---|---|
| R2 | Train | 0.~~7501~~9309 | 0.~~6003~~4514 | 0.~~8545~~9613 | 0.~~2931~~8380 | 0.~~3998~~3470 | 0.~~2707~~3775 | 0.~~8003~~8610 | 0.~~3061~~3382 | 0.382~~65510~~ |
| | Test | 0.8092 | 0.5822 | 0.8841 | 0.5190 | 0.5587 | 0.2744 | 0.8435 | 0.3037 | 0.4970 |
| MAE | ~~286.5762~~Train | ~~448.4605~~132.7429 | ~~245.1695~~407.0814 | ~~613.534~~275.1250 | ~~483.3066~~308.9700 | ~~615.3747~~410.2945 | ~~366.4148~~502.0053 | ~~543.4567~~236.9516 | ~~453.3241~~470.1633 | 276.20000 |
| | Test | 209.8063 | 435.5836 | 146.6120 | 510.6015 | 443.2222 | 614.3769 | 330.4876 | 536.0343 | 376.6252 |
| RMSE | Train | 348.6190 | 940.4850 | 113.4940 | 570.0070 | 1027.3730 | 1001.7620 | 574.9720 | 1042.9110 | 916.5471 |
| | Test | 646.5438 | 1047.4880 | 501.8960 | 1055.9190 | 1115.5270 | 1234.1220 | 737.0857 | 1237.9420 | 1176.9410 |
| MAPE | Train | 0.5240 | 0.7930 | 0.1540 | 76.3530 | 0.6280 | 5.2310 | 0.3810 | 1.5330 | 1.1588 |
| | Test | 0.5623 | 0.8892 | 0.3132 | 1819.2220 | 0.6623 | 4.1277 | 0.4939 | 5.8428 | 1.0421 |
| SMAPE | Train | 43.8375 | 79.8680 | 13.1780 | 150.4262 | 67.4715 | 103.0555 | 52.3359 | 93.4002 | 67.3221 |
| | Test | 39.7998 | 81.4539 | 22.7237 | 152.4991 | 73.6498 | 106.9756 | 63.7582 | 93.9244 | 76.9794 |


To dive deep into the prediction performance of the EGB model, we analyzed variables
importance in the prediction of the volume. It was observed that ~~the~~ slope length was the most
contributing predictor in the performance of the EGB model, followed by ~~the~~ maximum hourly
rainfall and slope aspect. The ~~presence~~altitude, three hours rainfall, slope angle and ~~quality~~age of
~~drainage ranked the third most contributor~~timber contributed moderately in the prediction of the
~~volume of rainfall due~~outcome volumes with gain above 0.01 and less than 0.2. the antecedent
rainfall from three days and above and continuous rainfall had a minor contribution, with a gain
of less than 0.01 for each. The presence of rainwater drainage channels had a moderate contribution,
with a gain close to ~~landslides. In addition, age of timber (age~~0.01. On the other hand, the
contribution of ~~trees~~soil depth and forest density in the models was insignificant and far below
0.01. Though Figure 2(g) depicted the association between larger volumes and fire history, the
variable importance indicates that ~~were planted~~the relation was not significant. Even though some
variables had minor contributions, depending on the ~~area that faced landslides) and maximum~~
~~hourly rainfall have~~case, the contribution of those variables may also ~~shown a significant~~
~~contribution in the prediction of volume of landslide due to rainfall. Figure 5~~increase depending
on other regional settings. Therefore, all variables with Generalized variance inflation factors
below 10 were kept in the model. Figure 6 illustrates ~~a list of independent variables that had a~~

significant impact in the prediction of the volume the variables importance for the EGB model. The vertical red line split the variables into two groups, the first containing variables that contributed a gain above 0.01 and others with minor contributions.

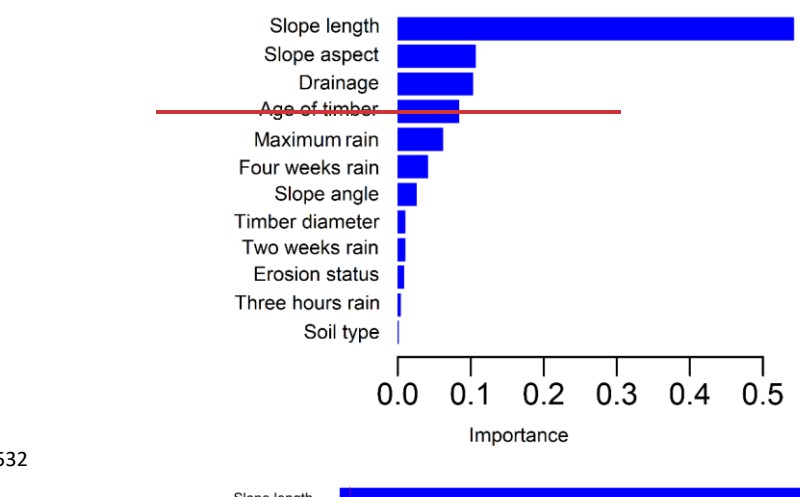

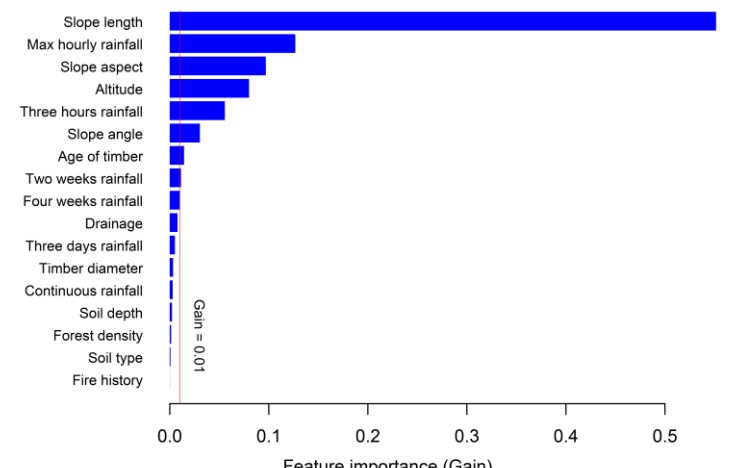

Figure 56. Variable importance for the EGB model.

The variable importance plot depicts the overall contribution of a given variable; however, it does not provide detailed information. To get more insight into the relationship between the volume of landslides and predictors, statistical tests for normality, namely, Shapiro-Wilk's test,

~~Kruskal Wallis test,~~ and Dunn's test were conducted. The Shapiro-Wilk's test (Dudley, 2023)
results revealed that the distribution of volume was non-normal (W = 0.40642, p-value < 0.001).
Noting that the volume distribution was non-normal, we opted for the non-parametric tests,
which do not rely on normality to conduct the distance correlation (Székely et al., 2007) test (dcor)
for continuous independent features. Figure ~~6~~7 illustrates that the slope length exhibited a higher
value (dcor=0.~~51~~56) followed by continuous rainfall ~~features. This highlights the role of~~
~~current~~altitude and ~~antecedent~~three hours rainfall ~~as triggering factor in~~and kept decreasing up to
timber diameter with a distance correlation of 0.08. Overall, the distance correlation between the
~~prediction of~~ volume of landslides shows a moderate strength of association between continuous
predictors.

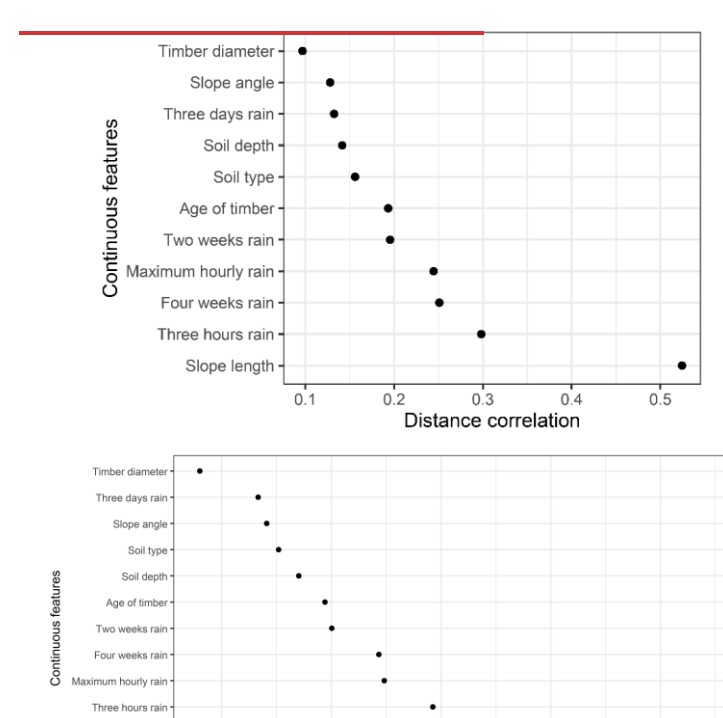



Figure 67. Distance correlation plot for the volume and continuous features.

Furthermore, to test for categorical features, Kruskal-Wallis test (McKight and Najab, 2010) was used to check whether the volume of the landslide was different in each category and Dunn's tests (Dinno, 2015) were applied to examine which categories had similar means of the volume of landslides due to rainfall in different categories. The $H_0$ (null hypothesis) was that the mean volume of landslides in different categories is the same, and the $H_1$ (alternative hypothesis) was that the means of landsides are different in some categories. For the slope aspect, the second most significant predictor for the EGB model, the results of Kruskal-Wallis test (chi-squared = 20.889, df = 7, p-value = 0.003938) showed that there is a significant difference in median of volume in some categories of slope aspects. To know which classes of slope aspects had significantly different mean volumes, the Dunn's test results at 95% confidence interval, pairs (East-South west, East-South East, East-South, East-North West and North West-South East) had significantly different means of landslides' volume (with p-value <0.05). Figure 78 depicts that the southwest and southeast aspects had a higher frequency of landslides.

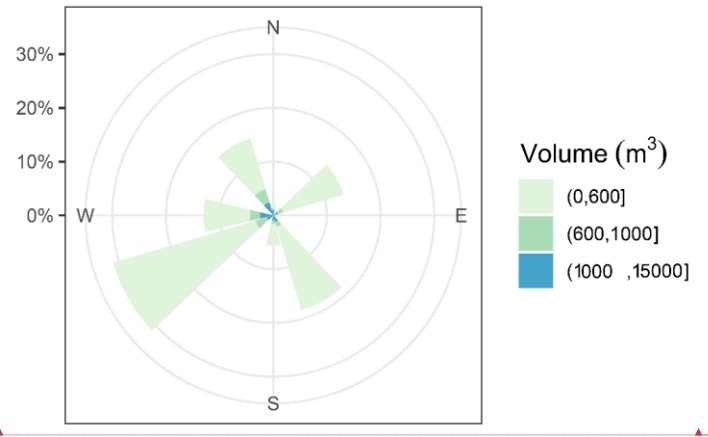

Figure 78. The distribution of the volume of landslides due to rainfall with respect to the slope aspect.

The Kruskal-Wallis test for the difference in mean of drainage classes showed the result was: chi-squared = 15.792, df = 2, p-value = 0.000372, which shows that the means of volume per classesclass were different. This was clarified by Dunn's test results, were p-values were less than

0.05 in all pairwise mean difference comparisons. The results of these tests highlighted that the drainage has a remarkable influence on the occurrence of rainfall-induced landslides in the Korean Peninsula.

## 5. Discussion

This study aim was to construct data a data-driven algorithm that predict predicts the volume of landslide landslides due to rainfall. The result of nine different tested algorithms revealed a tremendous difference between classical regression models (OLS, RR, and GLM) and other data-driven machine learning models. In this study, apart from SVM regression, DT and KNN, other machine learning models (DNN, DT, RF, and EGB) exhibited high prediction capability with $R^2$ above 50% (Fig.35). The random forest model performed well in predicting smaller volume however as the volume increased the model underpredicted volume values. The DNN model performed quite well with low MAE compare compared to random forest; however, the model did not perform on well on moderate volume values which resulted, resulting in reduction of reduced $R^2$. The EGB model tested on South Korean landslide inventory coupled with rainfall data at the time of landslide events and antecedent rainfall within one month of the event exhibited the highest performance compared to other constructed algorithms. The difference in performance may be due to the internal structure of each algorithm; the RF build multiple decision trees and averages predictions to improve accuracy (Breiman, 2001), while the EGB builds sequential trees in a recursive order where the new built tree improves error occurred while building the previous decision tree and optimizes the loss function through a gradient descent (Chen and Guestrin, 2016).

The slope aspect played an important role in the prediction of the volume, and the landslide mostly occurred on location in locations oriented toward south west southwest and south east southeast. That may be due to the direction taken by typhoon typhoons, which hit the south west southwest versants of mountains upon landfall on the Korean peninsula toward the North East Pacific (Ha, 2022; Lee et al., 2013). The findings of this research are congruent with those of Lee et al. (2013)), who also highlighted that the mountain versant oriented to strong wind direction may face more landslides. The study also highlighted that the efficacy of a moderate rainwater drainage channel plays an important role in the prevention of landslides which due to the its stabilizing effect. The landslide location and pattern follow the rainfall climate scenario, which highlighted a higher intensity of rainfall in the northeastern region of South Korea (Lee, 2016).

The findings of this study are congruent with Zhang et al. (2019) observations that
highlighted the low influence of soil type in landslide modeling and the maximum rainfall and
cumulative three hours of rainfall were the most contributing rainfall, which indicated that these
shallow landslides may have been triggered by sudden rainfall concentrated in few hours before
the occurrence of the event. The occurrence of landslides triggered by rainfall is a complex
phenomenon ~~which involve~~that involves many interrelated environmental ~~setting~~settings, human
activity, geological conditions and climatic conditions. Moreover, the occurrence of typhoons is
known to aggravate the landslides impacts on communities (Chang et al., 2008~~),~~); incorporating
typhoon variables in future studies to customize for regional ~~setting~~settings may improve the
accuracy of the model. The advantage of his research is that the constructed model has high
predictive accuracy and can handle the non-linearity of predisposing factors. The model came to
fill the gap of few literatures related to the prediction of the volume of landslides using data-driven
techniques. This model can be a ~~better~~good tool to help policy makers to integrate the landslides
volume risks in in policy to protect infrastructure and inhabitants dwelling near foot of mountains
with high risks of being buried by geological materials resulting from landslides.
To understand the applicability of the developed models, the trained model was tested using
unknown data (test data), with volume predictions generated solely based on the predictor
variables; actual volume values were utilized only for evaluating model performance. We found
that the DNN, EGB, GLM, and RF models achieved $R^2 > 0.8$, indicating that the model could yield
reliable volume estimates in adjacent areas with similar geological and environmental conditions.
It is also noted that the EGB, RF, and DNN are designed to efficiently handle large datasets, making
them suitable for regional-scale analysis with high scalability. Thus, these models can be scaled to
incorporate more data from different geographical areas without significant adjustments,
enhancing their applicability in future research (Krizhevsky et al., 2012). Subsequently, the
optimized model can aid in disaster risk management by providing timely information for early
warning systems. Additionally, the insights gained from the model can inform land-use planning
and policy decisions, allowing stakeholders to identify high-risk areas and implement mitigation
strategies effectively. By integrating the model into existing monitoring frameworks, agencies can
enhance their response capabilities and better allocate resources during heavy rainfall events.
The major limitation of this study is that the analysis is solely focused on shallow-seated
landslides, specifically translational slope failures with volumes below 13,000m³. Thus, the

analysis may not fully capture the variability in landslide characteristics across different geomorphological and geological contexts. Deep-seated landslides, for instance, often exhibit distinct failure mechanisms, material compositions, and depositional patterns that influence their volumetric characteristics, which were not considered in this investigation. Similarly, debris flows, known for their unique channelization and entrainment behaviors, were not included, potentially limiting the applicability of the optimized models to other landslide types. Further, this study was also performed using point-based landslide inventory data, which may not capture all variability of influencing factors and their exact state. The incorporation of high-resolution data from remote sensing and other sources may also improve the efficiency of the predictions. These limitations may impact the broader applicability of the proposed model; however, future studies will aim to address this by conducting separate analyses for deep-seated landslides and debris flows, allowing for a more comprehensive understanding of landslide volume predictions across diverse landslide types and geomorphological settings.

## 6. Conclusions

In this paper, the aim was to construct a data-driven model that predicts the volume of landslides due to rainfall. To this, nine different classical regression models and machine learning algorithms were tested on South Korean landslide data set containing features of landslides that occurred between 2011 and 2012. Among the tested models, extreme gradient boosting (EGB) produced the most accurate prediction. This is proven by the evaluation of the difference between actual and predicted values, such as $R^2$ = 88.41% and MAE = 146.6120m$^3$ on the test set. The analysis of feature variables in the contribution to the prediction of the model revealed that the slope length was the most influencing predictor. The EGB model can be a promising tool for the prediction of the volume of landslides due to its high predictive performance. The model can be customized in different environmental settings. The model can be applied to estimate the expected volume of landslides based on forecasted rainfall once the model is well-adjusted to fit the geomorphological and environmental settings of the region of interest after re-training on the regional historical data to include regional variability. Therefore, this model can be a good tool for planning for resilience and infrastructure pre-construction risk assessment to ensure the new infrastructure is placed in stable regions free from severe landslides.

**Acknowledgments**

This research was supported by through the National Research Foundation of Korea (NRF) funded by the Ministry of Education and Ministry of Science and ICT (2021R1A6A1A03044326, 2021R1C1C2003316). The authors highly appreciate both anonymous reviewers for their constructive suggestions that helped us improve the earlier preprint version.

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
