# Peer review of "Prediction of volume of shallow landslides due to rainfall using data-driven models"

_Natural Hazards and Earth System Sciences, 2024_

## Referee Comment (RC1)

**General comments**:

The problem of landslide volume estimation has been a focus for the community for quite some time, through methods such as area-volume scaling, geometrical modelling, numerical simulations, and more. This parameter is crucial as it helps gauge the magnitude of landslides, particularly at regional scales. Most highly accurate methods, like numerical simulations, often struggle at the regional scale. This manuscript offers a valuable reflection of data-driven modelling for delivering robust regional-scale analyses of landslide masses. Kudos to the authors for this interesting research, which has significant implications for hazard prediction and modelling. However, there are some major comments and curiosities I have. I believe the study is promising and of great interest to the landslide community, but it requires further work. The English language writing can be improved, especially in the Introduction. Some sentences read awkwardly and are hard to follow. Sentence phrasing must be improved to make the manuscript clearer, particularly for non-native English readers.

**Specific major comments**:

1. The Introduction needs to be revisited for editing in both grammar and phrasing of the language. Moreover, the motivation for the importance of volume quantification appears to be a bit lacklustre. I do not see a geomorphological connection as to why volume estimates are important to understand process mechanism and kinematics. Although, the manuscript does not explore said mechanism and kinematics expressions, however, to build a succinct story, a logical connection between the geomorphology and the surface failure should, in my opinion, be expressed to highlight why volume estimations are important as it directly feeds into the story of hazard prediction moving forward.

2. Are the training and testing datasets split randomly with keeping the training data fixed or is the split performed geographically? It would be interesting to see a geographically split dataset to see how well the model(s) perform due to apparent differences in the geological and environmental conditions across the study area.

3. One of my main concerns, or rather my curiosity, is regarding the data set itself. The volume information, along with the inventory, is particularly noteworthy in this case, as most inventories lack volume data. Keeping this in mind, how do the authors think about the application of such methods in other areas? Now, the authors have created a method that works pretty well within the given region. Instead of finding other regions (which might be difficult and time-consuming) could the authors simply use the model and predict volumes on similar nearby regions where the volumes are not calculated? This could serve as a simple prediction example demonstrating the method's application, without requiring extensive investigation. This approach is important as it helps the authors extend beyond a simple 'exercise' of the method, since it is currently applied only in the study area. Moreover, this would make the claim in Conclusion, Lines 346-349 more credible.

4. My biggest concern is related to the soil-depth. Now, it is impossible to imagine the calculation of volumes without the depth of the material that has failed as that is the 3$^{rd}$

dimension required for volume calculations. It appears that the soil depth was 'removed' after feature importance analysis for the best performing EGB model. Sure, the depth information might not have been that important in this example of model training for this region, but I would argue that in other regions, particularly if the region contains multiple deep-seated landslides and the failure surface runs deep until the bedrock. I am just not convinced that removing soil depth makes sense, as geomorphologically, depth (which also relates to soil composition) is very important for accurate volume estimation and calculation.

5.  Another question is pertaining to the type of failure movement. The inventory contains multitude of information but what about the movement types of the landslides? What types of landslides are considered in the inventory? Because clearly shallow and deep landslides would require separate treatments when looking at volume predictions because the material composition, material type, and material depths would be tremendously different. Do the authors combine these landslides together? What is the proportion of these landslide types? Also, are there prevalent debris flows, because volumes of debris flows is another story altogether since entrained volumes due to channelization are different than surface failure volumes. I see that the Discussion can be improved a lot by addressing and discussing these topics and limitations.

6.  The Discussion section is oriented quite too much on the aspects of the different models, conditioning factors, and their roles in the prediction of the volumes. As I mentioned in my previous comment, not much is discussed on the practical questions of scalability, different modes of movements, soil depths, runout volumes of entrained materials etc. These are essential topics as the direct counterpart of statistical models, i.e., numerical models tend to answer these questions. So, a comparison with the literature in that order is missing which I believe would add new levels of arguments to put forward by the authors and cement why their method works well despite lacking/following physical laws.

7.  In Table 1, under Geomorphology, the feature "erosion" is presented. Now, erosion itself can be referred to the volume, which is the main variable that the authors are trying to estimate. So, how is this variable used in the training regime? Or is this erosion feature different than the output of volume? Also, there are summary statistics of the erosion under Table 2. Why is that? My concern is that the authors are not clear as to what 'erosion' refers to in the data-driven model construct. If it is in fact similar to volumes, then the predictor variable and output variables are more or less the same. This needs further in-dept clarification.

8.  Table 1: Descriptions should be written properly for each feature/variable. At the moment, the descriptions read more like a summary of the sub-groups, written altogether. Please provide descriptions individually for each feature properly. For example, Slope angle, slope aspect, and slope length are all written in one statement. Make them three individual statements to make it clearer to understand. Also, the descriptions are not clear enough. For

example, "There exists an established relationship between the slope morphology and volume landslide due to rainfall". This is not a description. It is a reasoning to justify a claim. Please provide appropriate descriptions.

9.  Lines 311-312: It would be nice explain why the random forest works well with smaller volumes. The connection between the machine learning predictions and the scale of the estimated volumes should be explained more intricately to provide a grounded understanding. Does the EGB model predict larger volumes more accurately than the rest, like Random Forest? If so, then why? Please explain these aspects.

**Minor comments**:

1.  Line 31: "high", should be "height".
2.  Line 36: "resulting volume of landslides". Change this to "resulting surface failure".
3.  Line 38: "fragilize". Not sure if such a word is used commonly to express the weakening of slopes. I'd rather opt for 'weaken'.
4.  Similar English issues are found in Section 2 (Study area). Please address the language issues.
5.  Figure 2. Font size of plot (b) is different than the rest, and also stretched. Please make all font sizes uniform.
6.  Line 111: Replace 'joined' with 'combined'.
7.  Line 128: "flown away"? I am not sure if using this term is accurate. Generally, we refer to them as "removed material" from the surface. Can you please double-check this?
8.  Is the slope angle the average angle of the terrain where the landslide was located or is the angle of reach? In my opinion, the angle of reach would make more sense as landslides that are closer to each other will exhibit different angles of reach but the same adjacent landslides would bear the same average slope angle as you are averaging based on the terrain. Please make it clear as to which one you have considered and why.
9.  Line 136: What do you mean by 'composing material'? This is not clear.
10. Lines 140-142: Please check the English grammar here. The sentence can be improved a lot.
11. Line 341: Change to "Among the tested models,"
12. Conclusion- Line 349: Change from "can be a better tool" to "can be a good tool".

---

## Author Response (AR1)

**Manuscript number: nhess-2024-90**

My co-authors and I would like to express our gratitude to the reviewer for his constructive feedback and suggestions for strengthening our research. The changes we have made to the attached file in response to such feedback and suggestions have been highlighted in blue to facilitate their identification. I would also like to offer my apologies for the length of time it took us to prepare this response. We also record our deep appreciation for the efficient handling of the manuscript.

**Response to Reviewer#1**

**General remarks:** I am attaching my full comments in the attached PDF. At the same time, I am summarizing my general comments here for the editor's perusal.

This manuscript presents a valuable reflection of data-driven modelling for robust regional-scale analyses of landslide masses. The authors deserve commendation for their interesting research, which has significant implications for hazard prediction and modelling. However, I have some major comments and concerns. While the study is promising and of great interest to the landslide community, it requires further work. Some aspects of the training and testing regimes are not clear. Furthermore, the choice of certain parameters is not well justified which, in my opinion, must be clarified for readers to understand the logic of choosing said parameters. The English language, particularly in the Introduction, needs improvement. Some sentences read awkwardly and are hard to follow. Improved sentence phrasing is necessary to make the manuscript clearer, especially for non-native English readers. In my opinion, a major revision is required to adapt the manuscript before considering acceptance.

Response: Thank you for your detailed comments and for the recognition of the value of our research. We appreciate your commendation and acknowledge the importance of addressing your highlighted concerns. In the revised manuscript, we have focused on the aspects of the training and testing datasets to enhance understanding, as well as provide a stronger justification regarding the choice of predictor variables to ensure the logic is clear to all readers. Additionally, we revised the language throughout the entire manuscript to enhanced readability.

**General comments:** The problem of landslide volume estimation has been a focus for the community for quite some time, through methods such as area-volume scaling, geometrical modelling, numerical simulations, and more. This parameter is crucial as it helps gauge the magnitude of landslides, particularly at regional scales. Most highly accurate methods, like numerical simulations, often struggle at the regional scale. This manuscript offers a valuable reflection of data-driven modelling for delivering robust regional-scale analyses of landslide masses. Kudos to the authors

for this interesting research, which has significant implications for hazard prediction and modelling. However, there are some major comments and curiosities I have. I believe the study is promising and of great interest to the landslide community, but it requires further work. The English language writing can be improved, especially in the Introduction. Some sentences read awkwardly and are hard to follow. Sentence phrasing must be improved to make the manuscript clearer, particularly for non-native English readers.

Response: We appreciate the thoughtful feedback and for recognizing the value of our research in the context of landslide volume prediction and acknowledgment of the challenges faced by highly accurate methods at regional scales, and we appreciate that our data-driven modeling approach resonates with the landslide community. We took your concerns regarding the clarity of the English language and improved the phrasing and overall readability, particularly in the Introduction, to ensure it is accessible to all readers.

**Specific major comments:**

Comment 1: The Introduction needs to be revisited for editing in both grammar and phrasing of the language. Moreover, the motivation for the importance of volume quantification appears to be a bit lacklustre. I do not see a geomorphological connection as to why volume estimates are important to understand process mechanism and kinematics. Although, the manuscript does not explore said mechanism and kinematics expressions, however, to build a succinct story, a logical connection between the geomorphology and the surface failure should, in my opinion, be expressed to highlight why volume estimations are important as it directly feeds into the story of hazard prediction moving forward.

Response: Thank you for your insightful comment. We appreciate your suggestion to improve both the grammar and phrasing to enhance clarity. We also acknowledge the need to strengthen the motivation for volume quantification and its geomorphological significance. In the revised manuscript, we emphasized the connection between volume estimates and the understanding of process mechanisms, illustrating their importance in the context of hazard prediction. The revised Introduction is given below,

[revised manuscript text omitted]

Comment 2: Are the training and testing datasets split randomly with keeping the training data fixed or is the split performed geographically? It would be interesting to see a geographically split dataset to see how well the model(s) perform due to apparent differences in the geological and environmental conditions across the study area.

Response: Thank you for your insightful suggestion, which helped us improve the manuscript. In the present study, we opted to split the training and testing data randomly, implementing a 10-fold cross-validation to obtain an optimal model. This choice was made to balance bias and

variance effectively, adhering to a common 70% training and 30% testing split frequently employed in machine learning models (Nguyen et al., 2021), which has been shown to be an optimal data ratio.

While a geographically-based split could offer insight into regional variability, it may introduce challenges for this study, as landslide occurrences in our dataset are unevenly distributed, with about 60% located in the northeast part of the country. Geographically splitting this region as the test set would significantly reduce test data size, which could compromise model reliability and result in a suboptimal training process. To address regional variability without introducing geographic splitting, we incorporated altitude as a predictor variable in the model, recognizing that orographic rainfall in higher-altitude regions impacts soil saturation and may influence landslide susceptibility differently across regions. This approach allows the model to account for environmental differences while maintaining a balanced and representative dataset.

Comment 3: One of my main concerns, or rather my curiosity, is regarding the data set itself. The volume information, along with the inventory, is particularly noteworthy in this case, as most inventories lack volume data. Keeping this in mind, how do the authors think about the application of such methods in other areas? Now, the authors have created a method that works pretty well within the given region. Instead of finding other regions (which might be difficult and time-consuming) could the authors simply use the model and predict volumes on similar nearby regions where the volumes are not calculated? This could serve as a simple prediction example demonstrating the method's application, without requiring extensive investigation. This approach is important as it helps the authors extend beyond a simple 'exercise' of the method, since it is currently applied only in the study area. Moreover, this would make the claim in Conclusion, Lines 346-349 more credible.

Response: Thank you for your insightful comment. We agree that extending the applicability of our model to other regions is a valuable goal. While a comprehensive analysis of other regions is beyond the scope of this study, we recognize the potential to apply our model to similar regions with similar geological and environmental conditions.

In the present investigation, we selected a test set treated as unknown data to the model, where volume predictions were based solely on predictor variables, and actual volume values were used only to evaluate model performance. Our results indicate that the DNN, EGB, GLM, and RF models performed well, achieving an $R^2>0.8$. This level of accuracy suggests that the

model could provide reliable volume estimates in adjacent areas with comparable input data. We have clarified this point in the revised manuscript to highlight the model's adaptability.

Comment 4: My biggest concern is related to the soil-depth. Now, it is impossible to imagine the calculation of volumes without the depth of the material that has failed as that is the 3$^{rd}$ volume calculations. It appears that the soil depth was 'removed' after feature importance analysis for the best performing EGB model. Sure, the depth information might not have been that important in this example of model training for this region, but I would argue that in other regions, particularly if the region contains multiple deep-seated landslides and the failure surface runs deep until the bedrock. I am just not convinced that removing soil depth makes sense, as geomorphologically, depth (which also relates to soil composition) is very important for accurate volume estimation and calculation.

Response: Thank you for the fruitful observation. We agree that soil depth is important in the prediction of the volume of landslides due to rainfall. In this study, the average topsoil depth was considered, and during the training process, the contribution was minor in the prediction of volumes and values below 0.01 were not shown even though those features remained in the model. To remove the confusion caused by the absence of those variables with less contribution on the variable importance plot and to acknowledge that those variables may be more significant in other regions, all variables used to train all models were shown in the updated manuscript. The updated figure with its caption in the revised manuscript is depicted below,

[Figure]

Figure 6. Variable importance for the EGB model.

Comment 5: Another question is pertaining to the type of failure movement. The inventory contains multitude of information but what about the movement types of the landslides? What types of landslides are considered in the inventory? Because clearly shallow and deep landslides would require separate treatments when looking at volume predictions because the material composition, material type, and material depths would be tremendously different. Do the authors combine these landslides together? What is the proportion of these landslide types? Also, are there prevalent debris flows, because volumes of debris flows is another story altogether since entrained volumes due to channelization are different than surface failure volumes. I see that the Discussion can be improved a lot by addressing and discussing these topics and limitations.

Response: We appreciate the reviewer's insightful comments. We agree that landslide movement types are critical for accurate volume predictions, as they exhibit distinct failure mechanisms, material properties, and depositional patterns. As the reviewer correctly noted, our initial dataset contained a variety of landslide types. Upon further examination, we identified that the majority of landslides in our study area were shallow, translational slope failures. Only one deep-seated landslide, with an approximate volume of 33,000 m³, was included in the inventory. As observed in prior studies, shallow translational slides are common in granite areas of Korea due to uniform weathering profiles, while metamorphic regions tend to experience larger debris flows due to steeper slopes and irregular weathering profiles (Kim and Chae, 2009). Kim et al. (2001) further noted that in north and northwest part of the country, most landslides are classified as debris flows, though their initiation points often exhibit characteristics of translational slides. Recognizing that shallow and deep-seated landslides exhibit different material properties, failure mechanisms, and volumetric characteristics, we have removed this deep-seated landslide from our analysis to ensure consistency and relevance to our study objectives. We have therefore focused our analysis on this dominant type, as it represents the primary landslide hazard in the region.

This manuscript contains exclusively shallow-seated landslides with volumes below 13,000m³ with topsoil depth varying between 0.2m and 1m. We have updated the methodology and analysis sections to clarify that our dataset only includes shallow-seated landslides. Additionally, the Discussion section now addresses this limitation, acknowledging that the exclusion of deep-seated landslides and debris flows may affect the generalizability of our findings to other landslide types. This improvement aligns with the study's focus on shallow landslides, allowing for a more accurate assessment of volume predictions within this specific

landslide type. We have also noted in the Discussion that future studies would benefit from separate analyses of deep-seated landslides and debris flows, given the unique volumetric and channelization characteristics of debris flows.

Comment 6: The Discussion section is oriented quite too much on the aspects of the different models, conditioning factors, and their roles in the prediction of the volumes. As I mentioned in my previous comment, not much is discussed on the practical questions of scalability, different modes of movements, soil depths, runout volumes of entrained materials etc. These are essential topics as the direct counterpart of statistical models, i.e., numerical models tend to answer these questions. So, a comparison with the literature in that order is missing which I believe would add new levels of arguments to put forward by the authors and cement why their method works well despite lacking/following physical laws.

Response: This study aim was to construct a data-driven algorithm that predicts the volume of landslides due to rainfall. The result of nine different tested algorithms revealed a tremendous difference between classical regression models (OLS, RR, and GLM) and other data-driven machine learning models. In this study, apart from SVM regression, DT and KNN, other machine learning models (DNN, DT, RF, and EGB) exhibited high prediction capability with $R^2$ above 50%. Further, to understand the applicability of the developed models, the trained model was tested using unknown data, with volume predictions generated solely based on the predictor variables; actual volume values were utilized only for evaluating model performance. We found that the DNN, EGB, GLM, and RF models achieved $R^2>0.8$, indicating that the model could yield reliable volume estimates in adjacent areas with similar geological and environmental conditions. It was noted that the numerical models and machine learning approach mostly used for the landslide volume estimation depend on landslide geometry (Leong and Cheng, 2022; Do et al., 2017; Shirzadi et al., 2017). As or our knowledge, none of the ML models used to predict volume of landslides using multiple predictors (such as, geological, topographical, geomorphological, soil, vegetation, and rainfall factors) on large scale. Therefore, the direct comparison with result of existing numerical and statistical models that solely depend on geometrical features of landslide (such as, surface area or runout length) is out of the scope of this investigation.

Comment 7: In Table 1, under Geomorphology, the feature "erosion" is presented. Now, erosion itself can be referred to the volume, which is the main variable that the authors are trying to estimate. So, how is this variable used in the training regime?

Or is this erosion feature different than the output of volume? Also, there are summary statistics of the erosion under Table 2. Why is that? My concern is that the authors are not clear as to what 'erosion' refers to in the data-driven model construct. If it is in fact similar to volumes, then the predictor variable and output variables are more or less the same. This needs further in-dept clarification.

Response: We appreciate the reviewer comment regarding the confusion originating from the use of 'erosion' as a predictor variable. We agreed that the term may have caused confusion.

In the preprint, the feature named 'erosion' was incorporated as a categorical variable with 'Yes' and 'No' values, indicating whether minor erosion events (such as gradual surface degradation due to wind or water) occurred prior to the landslide event. This differs from the volume variable, which is our dependent variable and represents the total mass of displaced material due to a landslide. Importantly, volume was not used as a predictor in the model; rather, it serves solely as the target output. To avoid ambiguity, we have removed the 'erosion' variable from the predictor variable list in Table 1 and accordingly updated Table 2 in the revised manuscript.

Table 2: Summary statistics continuous variables.

| Variable | units | N | Min | Mean | Median | Max | Std dev |
|---|---|---|---|---|---|---|---|
| Max Hourly rain | mm | 455 | 0 | 48 | 48 | 78 | 20 |
| Continuous rainfall | mm | 455 | 0 | 285 | 327 | 550 | 106 |
| Three hours rainfall | mm | 455 | 0 | 88 | 80 | 171 | 60 |
| Twelve Hours rainfall | mm | 455 | 0 | 150 | 99 | 447 | 95 |
| One day rainfall | mm | 455 | 0 | 202 | 162 | 538 | 112 |
| Three days rain | mm | 455 | 0 | 280 | 284 | 550 | 86 |
| Seven days rain | mm | 455 | 0.5 | 323 | 330 | 634 | 88 |
| Two weeks rain | mm | 455 | 0.5 | 385 | 400 | 663 | 90 |
| Three weeks rain | mm | 455 | 86 | 504 | 533 | 914 | 115 |
| Four weeks rain | mm | 455 | 108 | 587 | 561 | 1135 | 160 |
| Soil depth | m | 455 | 0.2 | 0.6 | 0.75 | .75 | 0.19 |
| Soil type | - | 455 | 1.5 | 1.6 | 1.5 | 1.7 | 0.087 |
| Timber diameter | m | 455 | 0.15 | 0.27 | 0.23 | 0.35 | 0.086 |
| Age of tree | Years | 455 | 10 | 34 | 35 | 60 | 14 |
| Slope length | m | 455 | 1.8 | 21 | 13 | 180 | 23 |
| Slope angle | Degree (o) | 455 | 10 | 34 | 34 | 65 | 7.9 |
| Altitude | m | 455 | 9 | 391 | 272 | 1324 | 273 |

Comment 8: Table 1: Descriptions should be written properly for each feature/variable. At the moment, the descriptions read more like a summary of the sub-groups, written

altogether. Please provide descriptions individually for each feature properly. For example, Slope angle, slope aspect, and slope length are all written in one statement. Make them three individual statements to make it clearer to understand. Also, the descriptions are not clear enough. For volume landslide due to rainfall". This is not a description. It is a reasoning to justify a claim. Please provide appropriate descriptions.

Response: Thank you for your comment. We have updated Table 1 to enhance clarity by providing separate descriptions for each feature. Each description is now specific to the individual feature, detailing its relevance to landslide volume estimation. While rainfall parameters, such as rainfall on the day of the event and rainfall in prior days were grouped, as they represent related precipitation metrics, all other features have been distinctly separated. The revised version of Table 1, with improved feature descriptions, is shown below.

Table 1. Landslide influencing and triggering factors.

[revised manuscript text omitted]

Comment 9: Lines 311-312: It would be nice explain why the random forest works well with smaller volumes. The connection between the machine learning predictions and the scale of the estimated volumes should be explained more intricately to provide a grounded understanding. Does the EGB model predict larger volumes more accurately than the rest, like Random Forest? If so, then why? Please explain these aspects.

Response: Thank you for your insightful comment. Random Forest tends to perform well with smaller volumes due to its ability to capture complex relationships and interactions in the data without overfitting. RF uses multiple decision trees as base models, builds each tree on a random subset of samples and features, and computes averages as predictions to get the final result (Breiman, 2001). The model's random sampling of both observations and features allows it to build diverse trees; this enhances the generalization capabilities, particularly when the dataset is small. This characteristic helps the RF to maintain accuracy by reducing variance. It was noticed that the difference between of $R^2$ on training and testing sets was small compared to other models.

In contrast, the EGB model may predict larger volumes more accurately because it employs an iterative process to improve predictions. It uses a decision tree as the base model and builds them sequentially in such a way that each new tree corrects prediction errors made by previous trees using gradient descent, allowing for fine-tuning of predictions over iterations and to minimize loss functions effectively (Chen and Guestrin, 2016). This iterative correction can capture complex patterns in larger datasets that may not be evident in smaller ones. The fact that the RF predictions are averages of multiple decision trees may cause the difference since predicting averages will be less than predictions produced sequentially (Sagi and Rokach, 2018).

Furthermore, as volume size increases, the relationships between features can become more intricate, and EGB's ability to handle these complexities may lead to superior performance in those scenarios. However, Random Forest remains advantageous when data is scarce because it is less prone to overfitting compared to some boosting methods, which may struggle with limited data. A clear understanding of these dynamics provides valuable insights into the varying performance of different models across different volume scales, emphasizing the importance of choosing the right algorithm based on dataset characteristics. This has been highlighted in the discussion section of the revised manuscript.

**Minor comments:**

Comment 1: Line 31: "high", should be "height".

Response: Thank you for your observation. The identified error has been corrected in the revised manuscript. The entire sentence has been modified as,
"Landslides due to rainfall are phenomena that dislocate a mass of soil from its natural position and slide downward along a slope due to gravity forces."

Comment 2: Line 36: "resulting volume of landslides". Change this to "resulting surface failure".

Response: We have made the modification in the revised manuscript, replacing "resulting volume of landslides" with "resulting surface failure" for improved clarity.

Response 3: Line 38: "fragilize". Not sure if such a word is used commonly to express the weakening of slopes. I'd rather opt for 'weaken'.

Response: Thank you for your suggestion. Accordingly, we have replaced "fragilize" with "weaken" in the revised manuscript.

Comment 4: Similar English issues are found in Section 2 (Study area). Please address the language issues.

Response: Thank you for the fruitful suggestion and observations. We have addressed the language issues throughout the entire manuscript, including the Study Area section. The modification made in the study area section is reflected in the text below:

"The region for testing the model is South Korea, characterized by mountainous (63% of total land) relief, especially in the eastern part of the country (Lee et al., 2022). South Korea is located on the southern part of the Korean Peninsula, bordered by the Yellow Sea to the west coast and the East Sea (Sea of Japan) to the East. According to the Korean Meteorological Administration (2020), the country has a temperate climate characterized by four distinct seasons: hot and humid summers, cold winters, and springs and falls with moderate temperatures. The annual rainfall ranges between 1000 mm to 1400mm and 1800mm for the central region and southern region, respectively (Jung et al., 2017; Alcantara and Ahn, 2020). During the summer, heavy rainfall from June to September leads to significant surface runoff, increases landslide risk, and causes approximately 95% of all landslides each year (Lee et al., 2020; Park and Lee, 2021). In addition, the landslides may be aggravated by typhoons, which mostly occur in August and September, and it is anticipated that frequency will increase due to climate change (Kim and Park, 2021). The rainfall trend analysis from 1971 to 2100 predicted the increase in rainfall of 271.23mm, which indicates the growing risk of landslides associated with climate change (Lee, 2016). Temperature variations are influenced by its geographical location, the average summer temperatures range between 25 and 30°C, while winter temperatures can drop to -10°C in some parts of the country (Korea Meteorological Administration, 2020). The South Korean geologically is mainly composed of granitic and metamorphic rocks, such as gneiss, schist, and granite, which influence the stability of the landscape (Jung et al., 2024). The geomorphology is characterized by rugged mountains, river valleys, and coastal plains, with the Taebaek Mountains running along the eastern edge (Kim et al., 2020). In addition, the influence of rainfall, environmental, geomorphology, and

geological factors increase the vulnerability to landslides across the country, especially in the northeastern mountainous region, as depicted in Figure 1.

The predominant soil types in South Korea include clay, sandy, and loamy soils, each with different characteristics affecting water infiltration, retention and erosion (Kang et al., 2022; Lee et al., 2023). Clay soils, being more stable, can become highly saturated, increasing landslide risk during heavy rains. On the other hand, sandy soils are more prone to shallow landslides due to fast saturation, leading to instability. Regions with steep topography and poorly consolidated soil (loose) are mostly at risk, especially after prolonged rainfalls (Kim et al., 2015).

Coastal areas are exposed to sea-level rise and coastal erosion, which can further complicate the landscape and increase landslide susceptibility. The combination of heavy summer rainfall, geological composition, and geomorphological factors makes South Korea particularly vulnerable to shallow landslides. Thus, continuous monitoring and research are vital to understand the complex interactions between climate, geology, soil types, and landslide occurrences in this region (Park, 2022). Understanding the combination of environmental, geological stability, and geomorphological features is crucial for developing effective disaster management strategies and enhancing public safety in landslide-prone areas. As climate change continues to impact rainfall patterns, South Korea faces ongoing challenges in mitigating landslide risks and protecting vulnerable communities."

Comment 5: Figure 2. Font size of plot (b) is different than the rest, and also stretched. Please make all font sizes uniform.

Response: Thank you for your suggested improvements. Figure 2 (now Figure 3), titled "Workflow for the Prediction of Volume of Landslide Due to Rainfall," has been revised to ensure uniform font sizes throughout the plot. The updated figure is provided below,

[Figure]

Figure 3. Workflow for the prediction of the volume of landslides due to rainfall.

Comment 6: Line 111: Replace 'joined' with 'combined'.

Response: Thank you for your comment. As suggested, the term "joined" has been replaced with "combined" in the revised manuscript.

Comment 7: Line 128: "flown away"? I am not sure if using this term is accurate. Generally, we refer to them as "removed material" from the surface. Can you please double-check this?

Response: Thank you for your valuable comment. The suggested modifications were incorporated in the revised manuscript as,
"The estimation of the volume of removed material by landslides is important as it helps to assess risks the estimated damage can cause down at the toe of the failed slope, such as blocking transportation network, burying crops or farmland, the damage-built environment near landslide risks area, and post-disaster recovery planning (Evans et al., 2007; Rotaru et al., 2007; Intrieri et al., 2019)."

Comment 8: Is the slope angle the average angle of the terrain where the landslide was located or is the angle of reach? In my opinion, the angle of reach would make more sense as landslides that are closer to each other will exhibit different angles of reach but the same adjacent landslides would bear the same average slope angle as you are

averaging based on the terrain. Please make it clear as to which one you have considered and why.

Response: Thank you for your comment. The slope angle referenced in the manuscript pertains to the average angle of the terrain at the landslide location. This measurement provides valuable insight into the overall steepness and geomorphic characteristics of the area, which are crucial factors influencing landslide susceptibility and risk modeling (Donnarumma et al., 2013). On the other hand, the angle of reach refers to the angle at which a landslide material travels after detaching from the slope, which is important for assessing mobility and potential impact (Corominas, 1996). However, this is a different metric and not the focus of our analysis. While the angle of reach considers the mobility of landslides, the average slope angle is critical for assessing the risk of landslide occurrence. We acknowledge your point regarding the differences in angle of reach among closely situated landslides, but in our study, the average slope angle is more relevant for evaluating landslide volume. We have clarified this distinction in the revised manuscript (Table 1) to ensure a better understanding.

Comment 9: Line 136: What do you mean by 'composing material'? This is not clear.

Response: Thank you for your insightful comment. The term "composing material" refers to soil composition properties, which significantly impact slope stability. These properties, including soil permeability indices, influence water infiltration and saturation levels, both of which are critical factors in landslide susceptibility (Chen et al., 2015a). The revised sentence is as follows,
"The slope stability depends on soil composition properties, including soil permeability indices that affect water infiltration and saturation level (Chen et al., 2015a)."

Comment 10: Lines 140-142: Please check the English grammar here. The sentence can be improved a lot.
Response: Thank you for your comment. We revised the sentence in the updated manuscript. Additionally, we conducted a thorough review of the manuscript to identify and correct similar issues throughout.

Comment 11: Line 341: Change to "Among the tested models,"

Response: Thank you for your comment. The sentence has been modified in the revised manuscript as,

"Among the tested models, extreme gradient boosting (EGB) produced the most accurate prediction."

Comment 12: Conclusion- Line 349: Change from "can be a better tool" to "can be a good tool".

Response: Thank you for your comment. As suggested, the sentence has been revised in the updated manuscript as,

[revised manuscript text omitted]

**Response to Reviewer #2**

**General remarks**

**Comment 1:** In the introduction, the authors should explain more about why volume estimations are crucial for understanding and managing landslide hazards.

Response: Thank you for your valuable comment. As suggested, we have revised the introduction to highlight the critical role of landslide volume estimations in understanding and managing landslide hazards. The revised section of the introduction is provided below.

"To estimate the volume of the soil mass displaceable subsequent to intensive rainfall, is essential to set appropriate mitigation strategies to reduce environmental degradation, infrastructure damage, casualties, and to establish post-disaster resilience policies to restore the socio-economic aspect of communities (Van et al., 2021; Alcántara-Ayala, 2021). This quantification of the volume of landslides due to rainfall (VLDR) is essential for effective risk management (Tacconi et al., 2020), emergency response, engineering design (Cheung, 2021), economic assessment and environmental protection (Alcántara-Ayala and Sassa, 2023). Firstly, to manage landslide risk effectively, the quantification of VLDR can be useful for updating hazard maps to reflect the scale of potential landslides in various regions to facilitate the identification of high-risk zones for monitoring and intervention. In addition, to develop mitigation strategies, such as land stabilization measures and land use planning, planners might put in place strict construction regulations in particular regions that are susceptible to landslides (Mateos et al., 2020). The accurate measurements of VLDR can be used to promote public awareness for safety measures and preparedness (Yang and Adler, 2008). Secondly, estimating precise VLDR is crucial for structural engineers to design a structure that can withstand extreme landslide events. Knowing the exact volume of displaceable material, an engineer can set robust stabilization solutions to prevent future occurrences (Dai and Lee, 2001). Moreover, the VLDR can help design the drainage system to manage water flow by controlling groundwater and surface runoff to mitigate landslide risks (Dikshit et al., 2019; Kim et al., 2014). Furthermore, to prepare for emergence responses such as resource allocation, evacuation planning, and search and rescue operations, accurate VLDR estimation is necessary to ensure efficient implementation (Fan et al., 2019). To allocate resources effectively, the volume data is needed to determine the expected number of personnel for evacuation, materials sufficient for cleaning up and recovery (Amatya, 2016; Yang and Adler, 2008; Spiker and Gori, 2003). Further, to establish environmental protection measures such as ecosystem impacts,

preservation of soil and water quality, and habitat restoration, the estimates of VLDR are essential (Pradhan et al., 2022; Li et al., 2022a; Barik et al., 2017).

To mitigate the economic impacts of landslides, the values of VLDR can be a basis for estimation of property damages, which is critical for settling insurance claims and assessment of financial impacts on communities and government to facilitate efficient budgeting for repairing damaged infrastructure and restoration of affected parts (Klimeš et al., 2017; Dai et al., 2002). The prediction of the VLDR can assist in long-term economic planning for landslide risk by creating disaster preparedness and recovery funds (Winter and Bromhead, 2012). The accurate estimation of the VLDR is an important key for designing strategies for resilience and planning for the protection of the inhabitants of a particular region with certain landslide risks subjected to a predicted quantity of rainfall (Conte et al., 2022). Consequently, for the safety of communities, the selection of infrastructure construction sites must be done in places with low landslide risks (Fan et al., 2017). Further, for the protection of crops, the farmland location, and other land use activities, accurate landslide prediction taking into account real root causes through the analysis of triggering and influencing factors, is crucial to achieve a durable landslide safety management system (Paudel et al., 2003; Lee, 2009; Fan et al., 2017; Chen et al.,2019; Dai et al., 2019; Alcántara-Ayala, 2021). "

**Comment 2:** The literature review section should be expanded to incorporate more recent studies on landslide volume prediction models, providing a comprehensive overview of the current state of research in this field.

Response: We appreciate this suggestion. We have done an extensive literature review to include recent studies on landslide volume prediction models, offering a more comprehensive overview of the current state of research in this field. Accordingly, the literature review section in the introduction has been updated as,

"The prediction of VLDR has gained the interest of many researchers to understand the mechanism and interaction between triggering and aggravating factors. Saito et al. (2014) studied the relationship between rainfall-triggered landslides to test whether the volume of landslides across Japan that occurred between 2001 and 2011 can be directly predicted from rainfall metrics. The findings revealed that larger landslides occurred when rainfall exceeded certain thresholds, but there were significant discrepancies between peaks of rainfall metrics and maximum landslide volumes, and the total rainfall was the suitable predictor of landslides.

Dai and Lee (2001) established the frequency-volume relation for landslides in Hong Kong and noticed that the relation for shallow landslides above 4m$^3$ followed the power law. The 12-hour rolling rainfall contributed most to the prediction of the volume of landslides. Ju et al. (2023) constructed an area-volume power law model for the estimation of the volume of landslides using high-resolution LiDAR data collected between 2010 and 2020 in Hong Kong. The aim was to estimate accurately the volume of landslides on small-scale landslides. The reliance on localized datasets limits the model's applicability in regions with different geological settings, and the model does not consider all variabilities of landslide characteristics. Razakova et al. (2020) calculated landslide volume using remote sensed data with the aim of assessing the efficiency of aerial photographs in environmental impact assessment and ground-based measurement. The study did not take into account the effect of vegetation and topography and only focused on a single landslide case, which may be a source of bias due to differences in soil composition and environmental factors. Hovius et al. (1997) analyzed multiple sets of aerial photos and frequency-magnitude relation for landslides in New Zealand. The finding pinpointed that the landslides frequency-magnitude followed power law and infrequent large magnitude contributed to the landscape change. The study also noticed the importance of soil composition in the size of the landslides. This work had a limitation due to the reliance on aerial photos only, which cannot provide accurate measurement in regions of dense forest, and the climatic conditions, which are landslide triggering factors, were not considered, and this may affect the generality of the findings. Guzzetti et al. (2008) applied statistical methods on regional landslide inventories and antecedent rainfall data ranging between 10 min to 35 days. The findings revealed that the slope angle and soil type significantly influence landslide volume estimates, and the rainfall intensity is more important than duration. Chatra et al., 2019) applied numerical methods to study the effect of rainfall duration and intensity on the generation of pore pressure in the soil; the finding revealed a higher instability in loose soil compared to medium soil slopes. The work only treated the interaction of soil and rainfall without considering the environmental factors and human activity, which might also influence mass failure. Recently, the application of GIS technologies has been increasing in the identification of regions susceptible to landslides (landslide zonation) (Chen and Zhang, 2021; Gutierrez-Martin, 2020; Li et al., 2022b). These methods are essential in emergency management because they provide a general overview of zones with a higher probability of landslide occurrence; however, they do not put emphasis on the determination of the approximate value of the volume of failing mass in relation to excessive rainfall events."

**Comment 3:** The study area section should be enhanced with more detailed information on landslide-triggering factors. Additionally, it would be beneficial to incorporate a figure showing representative rainfall characteristics prior to the recorded landslide events in different parts of the Korean Peninsula. This would help better understand the unique rainfall patterns of the region responsible for landslides.

Response: We appreciate this fruitful comment. As recommended, we have included a rainfall plot within the data subsection in the revised manuscript to ensure logical coherence. Furthermore, we have combined the study area and data into a single section with two subsections for improved clarity. The updated information on the study area and the figure illustrating the rainfall characteristics prior to the recorded landslide events are provided below,

[revised manuscript text omitted]

Comment 4: Figure 2 needs to be updated. In the predictor variables, the authors should clearly specify which factors are influencing factors and which are triggering factors.

Response: Thank you for this observation. We agreed with the reviewer that the workflow had some missing information. Accordingly, the workflow figure was updated to reflect the reviewer's comment, and the model training and testing part was restructured to make it clearer. The updated Figure 3 (previously Figure 2 in the pre-print) is provided below,

[Figure]

Figure 3. Workflow for the prediction of the volume of landslides due to rainfall.

Comment 5: A more detailed discussion of the input variables considered for volume prediction is required to better understand their roles as influencing and triggering factors. Additionally, the manuscript should provide further justification for the selection of these predictor variables.

Response: Thank you for your insightful comment. The details about the input variable are summarized in the data part of the manuscript, and Table 1 provides the reference justifying the reason for considering the stated feature as an input variable of the model. Accordingly, we have thoroughly revised section 2.2 (i.e., Data) in the revised manuscript as follows,

[revised manuscript text omitted]

Comment 6: I recommend providing clearer details on the geometry of the landslide inventory.

Response: Thank you for this recommendation. The landslide inventory provided by the Korea Forest Service (KFS) contains 455 landslide records (point locations) information from 2011 to 2012 within the triggering area. This dataset tabulates information on landslide geometry, such as runout length, width, depth, and volume of the affected area, along with geomorphological composition, vegetation, and antecedent rainfall prior to landslide events, which are integral to understanding the spatial extent and impact of each landslide. These geometric details have been incorporated into our analysis to represent landslide characteristics accurately. Accordingly, the data section has been revised in the updated manuscript as follows:

"The landslide inventory dataset contains 455 landslide record information from 2011 to 2012, collected from different locations in South Korea by Korean Forest Services. This dataset tabulates information on landslide geometry, such as runout length, width, depth, and volume of the affected area, along with geomorphological composition, vegetation, and antecedent rainfall prior to landslide events."

Comment 7: A brief discussion on why nine data-driven models were chosen is recommended in the methods section. While these models have become quite common, providing a rationale for their selection will help justify their use in the study.

Response: Thank you for your valuable feedback. The inclusion of model selection is essential to justify the selection basis. In the present study, we aimed at predicting the volume of landslides using models that minimize error with interpretability and scalability. Since one model can not have all properties at the same time, we decided to select some of the models with those properties. The OLS, GLM, and DT were selected for their high interpretability, which helps to understand the influence of individual features on predictions (Gelman, 2007; Breiman, 2017). On the other hand, the EGB, RF, SVM, RR, and KNN were chosen due to

their robust performance in capturing complex patterns in data, which is essential for accurate predictions of landslide volumes (Chen and Guestrin, 2016; Liaw and Wiener, 2002; Hastie et al., 2009). Additionally, taking into account that the model will be used for regional scale, which will require the use of big data, the EGB, RF, DNN are designed to efficiently handle large datasets, making them suitable for the regional scale analysis. These last models can be scaled to incorporate more data from different geographical areas without significant adjustments, enhancing their applicability in future research (Krizhevsky et al., 2012).

Comment 8: The authors mainly use MAE and $R^2$ for model validation. It is recommended to consider additional metrics commonly used in data-driven model evaluation. Relying solely on these two statistics may not comprehensively assess model performance.

Response: Thank you for your valuable suggestion. We have expanded the metrics used for model validation to provide a more comprehensive assessment of the model performance. In addition to Mean Absolute Error (MAE) and $R^2$, we have included additional metrics such as Root Mean Squared Error (RMSE), Mean Absolute Percentage Error (MAPE), and symmetric mean absolute percentage errors (SMAPE). These metrics will offer a broader perspective on the accuracy and reliability of our predictions. The updated section detailing the selected metrics is included in the revised manuscript as,

"The model performance evaluation is a process of quantifying the difference between the observed value not used in the modeling process and the predicted value by the model. Different metrics are applied depending on the type of task, whether it is a classification or a regression problem. Subsequently, the widely used evaluation metrics for regression models, namely, $R^2$, MAE, RMSE, MAPE and SMAPE, were utilized to evaluate the model performances. The metric formulae and evaluation criteria are summarized in Table 3.

**Table 3.** Model evaluation metrics.

| Metrics | Evaluation | Reference |
|---|---|---|
| $RMSE = \sqrt{\dfrac{1}{n}\sum_{i=1}^{n}(y_i - \hat{y}_i)^2}$ | • Measures the square root of the average squared differences between predicted and actual values.
 • Lower values indicate better model performance. | Hyndman and Koehler, 2006. |

| | | |
|---|---|---|
| $MAE = \frac{1}{n}\sum_{i=1}^{n}\|y_i - \hat{y}_i\|$ | • The average of the absolute differences between predicted and actual values.
• Lower values indicate better model performance. | Willmott and Matsuura, 2005. |
| $MAPE = \frac{100}{n}\sum_{i=1}^{n}\left\|\frac{y_i - \hat{y}_i}{y_i}\right\|$ | • Measures the accuracy of a model as a percentage, which can be more interpretable.
• Lower values indicate better model performance. | Armstrong, 2001. |
| $SMAPE$
$= \frac{100}{n}\sum_{i=1}^{n}\frac{\|y_i - \hat{y}_i\|}{\|y_i\| - \|\widehat{y}_i\|}$ | • Unlike MAPE, which can be skewed by very small actual values, SMAPE accounts for both the actual and predicted values, making it symmetric.
• SMAPE is expressed as a percentage
• Mitigates the impact of small actual values on the error metric, providing a more balanced assessment.
• Lower values indicate better model performance. | Hyndman and Koehler, 2006 |
| $R^2 = 1 - \frac{\sum_{i=1}^{n}(y_i - \hat{y}_i)^2}{\sum_{i=1}^{n}(y_i - \bar{y})^2}$ | • Represents the proportion of variance in the dependent variable that can be explained by the independent variables.
• Values closer to 1 indicate a better fit | Darlington, 1990;
Chicco et al., 2021 |

\*$y_i$ and $\hat{y}_i$ representing the actual and predicted value and, $\bar{y}$ and $n$ standing for the mean of actual value and number of observations in the dataset, respectively.

Comment 9: The summary of the various data-driven models (Table 3) indicates that the EGB model is the best-performing. However, the variable importance analysis shown in Figure 5 highlights only a subset of predictor variables, raising questions about whether different models utilize different sets of features. Further clarification is needed.

Response: We appreciate your insightful comment regarding the inconsistencies between the summary of the various data-driven models in Table 3 (now Table 4) and the variable importance analysis presented in Figure 5 (now Figure 6). In the earlier version of the manuscript, the variable importance features with a value of gain below 0.01 were removed from the plot. To avoid those inconsistencies and make clear all variables, the figure was updated and reflect all variables used in the updated manuscript depicted below,

[Figure]

Figure 6. Variable importance for the EGB model.

**Specific Comments**

Comment 1: Figure 1(b): The y-axis label is missing.

Response: We appreciate this observation that helped us to improve the clarity of Fig 1(b), which was missing the y-axis. We have corrected this oversight, and the updated Figure 1 now includes the y-axis label. Figure 1(b) illustrates the boxplot of various rainfall features utilized in the model. The revised figure is provided below,

[Figure]

Figure 1. (a) Spatial distribution of landslides in South Korea, (b) temporal variation of rainfall, i.e., A: Maximum hourly rainfall, B: Four weeks rainfall, C: Three hours rainfall, D: Three days rainfall and E: Two weeks rainfall, (c) cumulative frequency distribution of volume of landslides and (d) box plot of volume of landslides.

Comment 2: In Figure 2, it would be better to use the terms 'Training and Testing Algorithms' instead of 'Run and Test Algorithms'. This terminology more accurately reflects the standard processes involved in model development.

Response: Thank you for your insightful observation. Figure 2 (now Figure 3) has been updated in the revised manuscript as follows,

[Figure]

Figure 3. Workflow for the prediction of volume of landslides due to rainfall.

Comment 3: Line No. 104-107: I recommend that the authors use the acronyms for the different data-driven models here, as they have already been defined earlier in the manuscript. Consistent use of these acronyms throughout the manuscript will improve clarity and readability.

Response: We appreciate the comment regarding the consistency in the use of acronyms in the manuscript to keep the clarity. The suggested consistency in the use of acronyms was adopted in the updated manuscript.

Comment 4: Line No. 150-152: 'Thus, planting vegetation is recommended as a better practice to improve soil cohesion and prevent potential landslides due to soil root interaction (Gong et al., 2017; Phillips et al., 2021)'. This is a recommendation, not a description. Please provide appropriate descriptions.

Response: Thank you for your comment. We understand the misuse of English connectives. The sentence was modified in the revised manuscript as follows,

"The absence of vegetation allows rainwater to seep away fine topsoil, causing shallow landslides (Gonzalez-Ollauri and Mickovski, 2017). On the contrary, vegetation improves soil cohesion and prevents potential shallow landslides due to soil-root interaction (Gong et al., 2021; Phillips et al., 2021)."

Comment 5: Why did the authors use 70% of the inventory and 30% for the validation? Why not 50% for each? The authors should state in the methods section why they used these percentages. Further, the authors need to clarify whether training and testing data were chosen randomly or if any specific criteria were used for the analysis.

Response: Thank you for your insightful suggestion. The training and testing were split randomly to train the model. The 10-fold cross-validation was performed to obtain an optimal model. This division aligns with common practices in machine learning, where a 70 (training): 30 (test) ratio is frequently used to ensure adequate training data while reserving a sufficient amount for model testing. In addition, Nguyen et al. (2021) also highlighted that 0.7/0.3 is the optimal split for the data.

Comment 6: Please review the references cited in the text, as there are frequent errors with the use of commas and semicolons between references. This issue occurs multiple times throughout the manuscript and needs correction for proper citation formatting.

Response: We thank you for pointing out the mis-references in the manuscript. We understand the concern regarding improper citations, and we carefully read and corrected all misreferencing.

Comment 7: Chen et al. (2015) is not cited correctly in the text. There are two different articles by Chen et al. (2015) listed in the references section. The authors need to distinguish between these references by specifying them as Chen et al. (2015a) and Chen et al. (2015b). The reference section and citation in the text should be updated accordingly to reflect these distinctions.

Response: We appreciate the observation regarding the two authors with the same name, and the correction has been done accordingly, as follows,

Chen, Z., Luo, R., Huang, Z., Tu, W., Chen, J., Li, W., ... and Ai, Y. (2015a). Effects of different backfill soils on artificial soil quality for cut slope revegetation: Soil structure, soil erosion, moisture retention and soil C stock. Ecological engineering, 83, 5-12.
Chen, T., He, T., Benesty, M., Khotilovich, V., Tang, Y., Cho, H., ... and Zhou, T. (2015b). Xgboost: extreme gradient boosting. R package version 0.4-2, 1(4), 1-4.

Comment 8: Line 133 cites (Kafle, 2022), but this article is either missing from the reference

section or is not cited correctly. Please verify and ensure that this reference is properly included and formatted in the reference section.

Response: The citation was mis-referenced. The citation has been corrected to "Kafle et al. (2022)" and is now properly included and formatted in the reference section of the revised manuscript.

Kafle, L., Xu, W. J., Zeng, S. Y., & Nagel, T. (2022). A numerical investigation of slope stability influenced by the combined effects of reservoir water level fluctuations and precipitation: A case study of the Bianjiazhai landslide in China. Engineering Geology, 297, 106508.

Comment 9: Line no 215: Chowdhury (2023)- article not present in the reference section or not mentioned in the correct form.

Response: Thank you for your comment. The reference has been updated to "Chowdhury et al. (2023)" in the revised manuscript, and it is now correctly included in the reference section as follows,

Chowdhury, M. Z. I., Leung, A. A., Walker, R. L., Sikdar, K. C., O'Beirne, M., Quan, H., and Turin, T. C. (2023). A comparison of machine learning algorithms and traditional regression-based statistical modeling for predicting hypertension incidence in a Canadian population. Scientific Reports, 13(1), 13.

Comment 10: Line no 236: (Team, 2022)- article is not present in the reference section or not mentioned in the correct form.

Response: Thank you for pointing out the citation issue. The reference has been corrected to "R Core Team (2022)" in the revised manuscript. The updated reference is provided below:

"R core Team (2022). R: A language and environment for statistical computing. R Foundation for Statistical Computing, Vienna, Austria. URL: <https://www.R-project.org/>."

Comment 11: Line no 239: Jerome et al. (2012)- article not present in the reference section or not mentioned in the correct form.

Response: We appreciate your observation regarding the mis-reference in line 239 of the

manuscript concerning Jerome et al. (2012). The reference was updated as "Jerome et al. (2010)" in the updated manuscript. The updated citation is as follows,

Friedman, J. H., Hastie, T., & Tibshirani, R. (2010). Regularization paths for generalized linear models via coordinate descent. Journal of statistical software, 33, 1-22. URL:<https://www.jstatsoft.org/v33/i01/>."

Comment 12: Please verify the unit of soil depth. Most landslide inventories presented in Table 2 and Figure 1(d) suggest that the landslides are shallow-seated based on their volume distributions; the unit of soil depth does not seem to align with this observation.

Response: We thank the reviewer for highlighting the mistake in the soil depth unit. The correct unit for soil depth is centimeters (cm). However, for consistency and clarity, we have transformed the soil depth data into meters (m) and updated the manuscript accordingly.

Comment 13: I suggest adding a few lines in the discussion section to highlight the practical applicability of the proposed model. This would provide insight into how the model can be used in real-world scenarios and its potential impact on practice or policy.

Response: Thank you for your suggestion regarding the practical applicability of our proposed model. In the revised discussion section, we included a few lines highlighting how the model can be utilized in real-world scenarios.

[revised manuscript text omitted]

---

## Author Response (AR2)

My co-authors and I would like to express our gratitude to the reviewer for his constructive feedback and suggestions for strengthening our research. The changes we have made to the attached file in response to such feedback and suggestions have been highlighted in blue to facilitate their identification. I would also like to offer my apologies for the length of time it took us to prepare this response. We also record our deep appreciation for the efficient handling of the manuscript.

**Response to Editor**

**Overall Observations:** Referees who have reported on your initial submission have evaluated your manuscript again. However, their comments read rather contrasting: While referee #1 recommends publication of the paper as it is, referee #2 still has important concerns regarding the scientific context of your article in accordance to earlier comments mostly considering motivation of your research and discussion of the results including model transferability.

Looking at the current version of your manuscript, I think you thoroughly revised your paper, which is now not far from being publishable in NHESS. However, I agree with referee #2 that the presentation of your research still requires some improvements as detailed in the attached report, but I am not convinced that the paper necessarily needs another round of external peer-review. Based on this, I like to advise you to revise your paper considering the comments made by the referee and resubmit a new version of the article accompanied by a detailed point-per-point reply letter on the comments of the reviewer, and a version in track change mode highlighting the applied changes. After resubmission, the editor will review the article again.

**Response:** Thank you for your incomparable assistance during the review process of this manuscript. We are grateful for the constructive and insightful comments provided by both reviewers, which have significantly contributed to enhancing the quality and clarity of our work. We sincerely appreciate your recommendation for minor revision and have carefully addressed the comments provided by Reviewer #2, as well as the suggestions from the Editor. These revisions have been incorporated into the revised manuscript.

**Response to Reviwer#2**

**Overall Observations:** Thank you for your response, dear authors. I appreciate the revisions made after the previous round of reviews, but I believe the manuscript can still be further improved with additional revisions before it is ready for publication. I have presented my comments and suggestions in this iteration of the review, which I hope the authors will find helpful in enhancing the manuscript further.

Thank you for your valuable suggestions and guidance, which have greatly contributed to improving the earlier version of the manuscript. We acknowledge that the Introduction and Discussion sections required further refinement, and we have carefully revised each part in accordance with your recommendations.

**Comment 1A:** I will begin the second review and my line of questions starting from the Introduction section. In the last round of review, I raised a general concern about the lack of connection between volume estimation, geomorphological process understanding, and engineering solutions. Unfortunately, I still observe two major issues with the revised version:

It appears that the authors have addressed a wide range of topics, including engineering solutions for mitigation, risk assessment, financial compensation, and related aspects. While these are undoubtedly important, the way they are presented lacks a clear and cohesive narrative, making it difficult for readers to follow. As a reader, it feels like I am encountering a series of disconnected bullet points about various applications of volume information for landslides and their societal or biodiversity impacts, without a clear sense of purpose or direction in the text. Let's take this for example, "Firstly, to manage landslide risk effectively, the quantification of VLDR can be useful for updating hazard maps to reflect the scale of potential landslides in various regions to facilitate the identification of high-risk zones for monitoring and intervention." Now, normally such statements (especially in a review) is followed by a general explanation as to how the volume information can be used directly for the purposes of "updating hazard maps", for instance, by illustrating how these updated maps help prioritize areas for additional ground-based investigations, early warning system placements, or resource allocation for slope stabilization efforts. In other words, the statement should detail the direct linkage between volume quantification and subsequent practical steps that can be taken to mitigate landslide risk, rather than simply asserting that such a connection exists without any further elaboration.

This type of simple assertion is not the best for a reader to gauge what really is going on. Particularly, if there is no direct link between volumes and the respective impact. Other statements have the same 'linking' problem. Moreover, an equally big issue is that the new added paragraphs read the same to me. I do not gain any new information from the new text. The authors mention: "mitigation strategies, effective risk management, emergency response, public awareness on safety measures and preparedness, drainage system to control surface runoff, determining expected number of personnel for 'clean up' and recovery, establishing ecosystem impacts, habitat restoration, protection of crops and farmlands." Frankly, these topics are very diverse and complex, spanning multiple engineering, scientific, and social science disciplines. However, I see no clear connection to the manuscript's main narrative. Are the authors implying that their method can address all of these issues simply because it can accurately predict volumes? Does all of South Korea face these problems (more or less) equally? My point is that I cannot discern a clear, concise rationale or storyline explaining why landslide volume estimation is necessary. I recommend re-writing the two paragraphs related to the volumes and associated topic in the Introduction more carefully. Please keep the linkage direct and to the point, while citing some examples from the literature.

**Response:** Thank you for your insightful observations. We sincerely appreciate your detailed and constructive comments regarding the Introduction section. We agreed with the reviewer's suggestion that the material presented lacked coherence in the Introduction section. Accordingly, we have restructured the Introduction to maintain a clear and direct connection between landslide volume estimation and its engineering and scientific applications. The revised text eliminates broad and generalized statements, focusing instead on specific, practical linkages between volume estimation and its role in hazard assessment, risk mitigation, and resource allocation. The revised Introduction is provided below,

[revised manuscript text omitted]

**Comment 1B:** Additionally, the authors did not adequately integrate the volume estimations or predictions into a geomorphological context. This aspect is crucial, as it forms the crux for studies linking sediment transport, material mobilization, and sediment influx into river systems for example. Omitting this perspective is problematic since it averts a reader from understanding how the observed volumes relate to underlying geomorphological processes (including hillslope process evolution), ultimately limiting the usefulness and practicality of the study's findings for both scientific insight and practical applications in landscape management and hazard mitigation.

I believe the current structure of the Introduction is not optimal. I encourage the authors to take their time and thoroughly revise this section, especially the two paragraphs related to volumes. I suggest a comprehensive overhaul of the discussion on the importance of volumes, incorporating key works by Montgomery, Jaboyedoff, Korup, and van Westen to strengthen the narrative. Please take this opportunity to carefully revise the text such that the importance of volumes is clear and coherent from an application point of view, ranging from both engineering and geomorphological perspectives.

**Response:** Thank you for your observations. We understand the importance of incorporating the geomorphological aspect, and some literature was included in the revised manuscript accordingly. We reread the introduction and rewrote it considering your suggestions to improve the technicality, clarity and logical coherence of the section. The revised Introduction provides a clearer and more coherent discussion of the importance of landslide volume estimation from both engineering and geomorphological perspectives. The updated introduction is highlighted in our response to Comment#1A.

**Comment 2A.** Moving on to the next topic, I want to stress a bit more on the geographical split

testing argument and the application of the model to other locations/regions.

Let's start with the geographical split. The authors state that dividing the data by region would compromise model reliability due to the reduced size of the test set. While I partially agree, the authors also mention that they incorporated altitude as a predictor variable to reflect geographical diversity citing the influence of orographic rainfall on higher-altitude areas. This reasoning, however, may be oversimplified. Altitude is only one dimension of regional variability and may not fully capture the complexity of geographic differences in landslide susceptibility (and/or by proxy, volumes). Although incorporating altitude could help the model account for some variations associated with elevation, it cannot completely substitute for explicit geographic variability. Regionally distinct factors—such as geology, lithology, vegetation, land use, and soil types—may not be adequately represented by altitude alone. Relying solely on altitude as a proxy for regional variability implies oversimplifying the spatial heterogeneity inherent in landslide processes.

Please note that, while I do support the approach for splitting the data, I disagree with the notion that altitude alone is sufficient to capture the geographic diversity inherent in South Korea's varied landscapes. In my previous review, my suggestion was to consider performing or evaluating a spatial cross-validation, although a regular 10-fold cross-validation could also suffice (as the authors noted that 60% of the data is concentrated in the northeast), despite the methodological differences between the two approaches. I am interested to hear the authors' thoughts on the use of altitude as a proxy for geographic diversity.

**Response:** Thank you for your insightful comment. We agree that relying solely on altitude to capture the geographic variability inherent in South Korea's diverse landscapes would be an oversimplification. In the present study, we incorporated additional variables in the modeling process, including soil types, soil depth, slope aspect (versant), drainage, and vegetation-related variables to capture the spatial heterogeneity inherent in landslide processes. Soil types and their coefficients of permeability reflect regional lithological variations; the drainage significantly affects slope stability and promotes efficient control of rainfall's influence on groundwater fluctuation; slope aspect accounts for potential influences of rainfall patterns and wind direction on slope vulnerability, and vegetation-related variables (age of tree, forest density, timber diameter) improve soil cohesion and prevent direct contact of raindrops with the soil surface as highlighted in the feature importance (Fig. 6). These selected variables contribute to capturing regional variability and improving the model's predictive

capability. We believe that the selected features provide sufficient insight on the application of suggested methods in the prediction of VLDR.

[Figure]

Figure 6. Variable importance for the EGB model.

**Comment 2B:** For the application of the model, the authors mention using an unknown dataset—presumably the independent test set—where the model achieved an R² value above 0.8, which is quite good. My previous recommendation was to see if the authors could apply the model elsewhere, ideally in a nearby area (still in the South Korean Peninsula) with new (or even old) landslides for which no volume information is currently available. While ML/DL models often perform well on familiar data, they may produce unpredictable, random or less meaningful results when applied in a 'new' region or context. This way, the authors might see how the model behaves under different conditions, while providing insights into its generalizability and practical applicability to scenarios beyond the training environment. Of course, the authors will not be able to validate these results (for N number of landslides) since no ground truth exist, but it will give a good idea if the predicted volume prediction numbers are off the charts (e.g., extremely large or very small). This is important to investigate how random the model(s)' predictions can be and, beyond that, provides additional motivation for the authors' work, moving it beyond merely a 'modelling exercise'.

**Response:** Thank you for your suggestion. We acknowledge the importance of testing the model in diverse conditions to evaluate its robustness and behavior under different

environmental and geological contexts. In the present study, to understand the applicability of the developed models, the trained model was tested using unknown data (test data), with volume predictions generated solely based on the predictor variables; actual volume values were utilized only for evaluating model prediction accuracy. The outcome exhibited that the difference in $R^2$ on the training and holdout set of 7.72% for the optimal model (i.e., EGB) highlights that the model can be applied to another region of a similar setting. It was noted that without proper model calibration with the independent data set, it's difficult to determine whether these discrepancies in performance are due to model limitations or data differences in different regions (Huang et al., 2020). Therefore, in future work, we plan to develop an independent database based on collecting the extensive recent landslide geometry at different parts of the Korean Peninsula to improve the models further by calibrating region-specific parameters to ensure the transferability of the model to other regions.

**Comment 3:** Regarding the Discussion, the authors stated in their response: "direct comparison with result of existing numerical and statistical models that solely depend on geometrical features of landslide (such as, surface area or runout length) is out of the scope of this investigation". It seems that the authors may have misinterpreted my suggestions. The recommendation was not to perform a numerical comparison with other methods, such as statistical or numerical models, but rather to review the literature on such methods and highlight why the authors' approach is reliable.

This suggestion is particularly important because, as the authors themselves mentioned, no previous study has used such a multivariate predictor approach for volume predictions. Therefore, it is important to discuss and review this aspect as a huge chunk of the literature rely on, for instance, numerical and geometrical methods for volume estimations. Additionally, it is crucial to discuss and review related topics in common geomorphological research—such as sediment transport, landscape evolution, and material mobilization—since these processes rely heavily on volume data for quantification. This connects back to the Introduction section, which I previously noted requires an overhaul. Elements introduced in that section can be further expanded upon in the Discussion to emphasize potential applications of the proposed approach. While volume information is undeniably important, simply focusing on the influence of ML/DL on model performance significantly underestimates the broader implications that the Discussion section could address given the scope and nature of the study.

**Response:** Thank you for your insightful comment. Regarding the improvement in the Introduction and Discussion, the suggested modifications have been incorporated in the updated version of the discussion section as follows,

**Line Nos 457- 480:**

Numerical models have traditionally been employed due to their foundation in physical principles such as slope stability and hydrological dynamics (Glade et al., 2005). These models are valuable for understanding the underlying mechanisms of landslide processes but often face limitations when applied to regions with complex or heterogeneous terrain, as they require detailed, high-quality input data that may not always be available (Caine, 1980). In the same way, statistical models, which use historical rainfall and landslide data to establish correlations, can offer useful predictions of VLDR in regions with extensive historical records (Chung and Fabbri, 2003). However, these models may struggle to account for local variations in topography or rapidly changing weather patterns, limiting their general applicability. Additionally, ML techniques have shown significant promise in improving predictive accuracy at the regional level due to the capability of processing large, diverse datasets and capturing complex, non-linear relationships that traditional models might fail to capture (Pourghasemi and Rahmati, 2018). Further, ML models can adapt to regional variations and continuously improve as new data is introduced, offering a more flexible and dynamic approach to predict VLDR on a regional scale (Liu et al., 2021). Subsequently, the aim of this study was to construct a data-driven algorithm that accurately predicts the VLDR. The result of nine different tested algorithms revealed a tremendous difference between classical regression models (OLS, RR, and GLM) and other data-driven machine learning models. In this study, apart from SVM regression, DT and KNN, other machine learning models (DNN, DT, RF, and EGB) exhibited high prediction capability with $R^2$ above 50% (Fig. 5). The DNN, EGB, and RF models achieved $R^2>0.8$ on both training and test set with accuracy reduced $R^2$ by 1.75, 7.72, and 12.17% for RF, EGB and DNN respectively, on the holdout set, indicating that the model could yield reliable volume estimates in adjacent areas with similar geological and environmental conditions. The random forest model performed well in predicting smaller volume; however, as the volume increased, the model underpredicted volume values.

**Line Nos. 514-525:**

"To understand the applicability of the developed models, the trained model was tested using unknown data (test data), with volume predictions generated solely based on the predictor variables; actual volume values were utilized only for evaluating model prediction accuracy. The outcome exhibited that the difference in $R^2$ on the training and holdout set of 7.72% for the optimal model (i.e., EGB) highlights that the model can be applied to another region of a similar setting. It was noted that without proper model calibration with the independent data set, it's difficult to determine whether these discrepancies in performance are due to model limitations or data differences in different regions (Huang et al., 2020). Therefore, in future work, we plan to develop an independent database based on collecting the extensive recent landslide geometry at different parts of the Korean Peninsula to improve the models further by calibrating region-specific parameters to ensure the transferability of the model to other regions."

**Comment 4:** Table 1 column 'Descriptions' seem misleading. Descriptions should also include the definition of the variables, not just the 'influence' of the variable. For example, for Slope angle, there's no definition as to what it means, but rather a statement which explains the influence of the slope angle (e.g., slope at 20-30 degrees more vulnerable to landslides due to rainfall). This is not really a 'description'. I suggest either changing the column name or adding a definition first for each variable and then explaining their influence on landslides.

Also, based on line 287, it seems that there are only three types of soil, sandy loam, loam, and silt loam. Please add them in the table for soil types as well.

**Response:** Thank you for your observations, and we agree with your suggestion. Accordingly, the word 'description' was replaced by 'feature relevance' in the revised manuscript. Additionally, the feature relevance of three types of soil has been incorporated in Table 1. The revised text incorporated in Table 1 is given below.

'Soil types, namely, Sandy loam, silt loam and loam, with their coefficient of permeability 1.7, 1.65 and 1.5, respectively, retain water differently, leading to different saturation times. The soil with higher permeability tends to drain water more efficiently, making it less prone to saturation. In contrast, the soil with lower permeability, the pore pressure rapidly increases, which leads to shallow landslide initiation during intense rainfall events.'

**Comment 5.** The authors have provided a clear explanation of the feature importance for soil depth, and I appreciate their decision to retain it, as it is crucial for volume estimations. The authors noted that soil depth could play a more significant role in different regional settings with varying behaviors or responses, and I agree with this perspective. I have no further comments on this matter.

**Response:** Thank you for your insightful comment. We appreciate your earlier suggestions, which helped us refine and clearly articulate the rationale for including soil depth as a critical predictor variable in the model.

**Comment 6:** I appreciate the response and explanation regarding the differences between the Random Forest and EGB models in predicting smaller and larger volumes, respectively. Indeed, an iterative process like EGB, guided by gradient descent, is likely to capture the more intricate patterns associated with landslides generating large volumes. Similarly, the 'average' behaviour of the ensemble approach in Random Forest effectively accounts for the prediction of smaller volumes on average. I have no further comments on this matter.

**Response:** Thank you for the comment, which helped us to improve the interpretation regarding the differences in the predictions of different models.

**Comment 7:** The authors have also explained the landslide movement query very well, and I have no further questions in that regard.

**Response:** Thank you for your feedback. We appreciate your observations, which have greatly improved the clarity and quality of the manuscript. We are pleased that the explanation regarding the landslide movement met your expectations.

**Comment 8:** In conclusion, my impression of the technical aspects of the work is positive since much of the authors' clarifications addressed my concerns. However, the justification for the importance of volume information, its applications, and the future scope remains limited, which undersells the contribution of this study. I believe that an additional round of revisions would further enhance the manuscript, making it more accessible and impactful for a broader audience. I wish the authors good luck with their revisions.

**Response:** Thank you for the positive response regarding the technical aspect of the manuscript. The suggested improvements were incorporated into the revised manuscript.

**References**

[revised manuscript text omitted]